# On Inherent Adversarial Robustness of Active Vision Systems

**Amitangshu Mukherjee**  *mukher44@purdue.edu*
*Elmore Family School of Electrical and Computer Engineering*
*Purdue University*

**Timur Ibrayev**  *tibrayev@purdue.edu*
*Elmore Family School of Electrical and Computer Engineering*
*Purdue University*

**Kaushik Roy**  *kaushik@purdue.edu*
*Elmore Family School of Electrical and Computer Engineering*
*Purdue University*

**Reviewed on OpenReview:** *https://openreview.net/forum?id=iVV7IzI55V*

## Abstract

Deep Neural Networks (DNNs) are susceptible to adversarial inputs, such as imperceptible noise and naturally occurring challenging samples. This vulnerability likely arises from their passive, one-shot processing approach. In contrast, neuroscience suggests that human vision robustly identifies salient object features by actively switching between multiple fixation points **(saccades)** and processing surroundings with non-uniform resolution **(foveation)**. This information is processed via two pathways: the **dorsal** (where) and **ventral** (what) streams, which identify relevant input portions and discard irrelevant details. Building on this perspective, we outline a deep learning-based active dorsal-ventral vision system and adapt two prior methods, FALcon and GFNet, within this framework to evaluate their robustness. We conduct a comprehensive robustness analysis across three categories: adversarially crafted inputs evaluated under transfer attack scenarios, natural adversarial images, and foreground-distorted images. By learning from focused, downsampled glimpses at multiple distinct fixation points, these active methods significantly enhance the robustness of passive networks, achieving a **2-21%** increase in accuracy. This improvement is demonstrated against state-of-the-art transferable black-box attack. On ImageNet-A, a benchmark for naturally occurring hard samples, we show how distinct predictions from multiple fixation points yield performance gains of **1.5-2 times** for both CNN and Transformer based networks. Lastly, we qualitatively demonstrate how an active vision system aligns more closely with human perception for structurally distorted images. This alignment leads to more stable and resilient predictions, with lesser catastrophic mispredictions. In contrast, passive methods, which rely on single-shot learning and inference, often lack the necessary structural understanding. [1]

## 1 Introduction

The human visual perception system is one of the most sophisticated and robust vision systems in the animal kingdom (Ungerleider & Haxby, 1994; Goodale & Milner, 2004; Clark, 2013; Shao et al., 2024).It

---

[1]The code is available at GitHub.

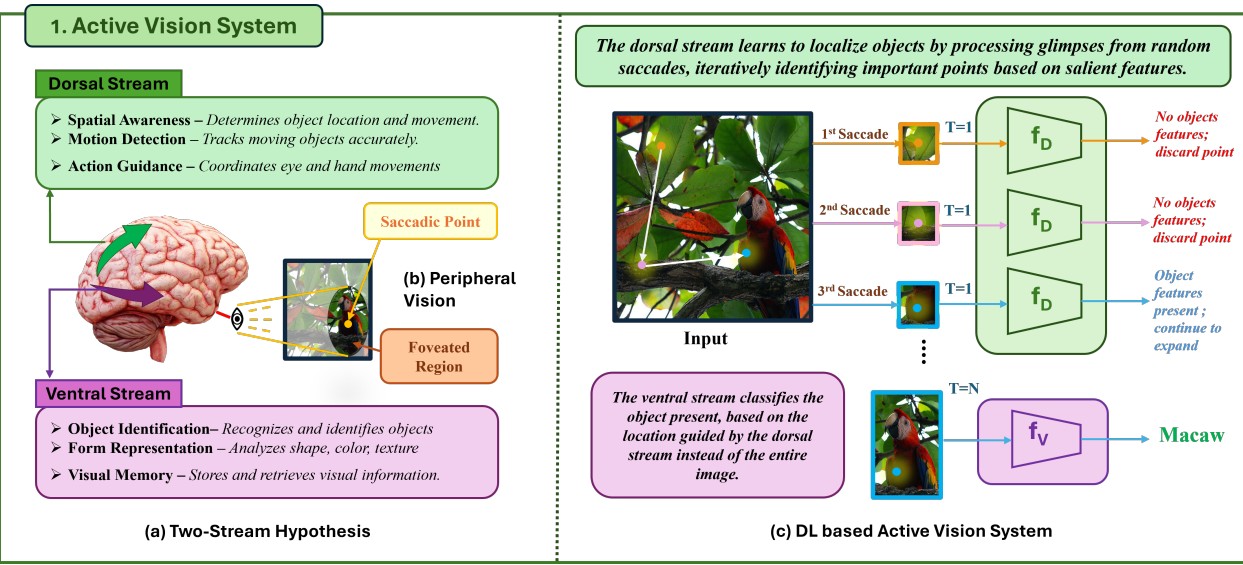

Figure 1: (1a) Illustrates the dual-stream hypothesis, highlighting the distinct dorsal ("where") and ventral ("what") pathways along with their key corresponding features. (1b) Depicts peripheral vision, emphasizing the inherent mechanisms of saccades and foveation. Together, the peripheral vision and the two-stream model form an active vision system. The corresponding DL-based active vision system is schematically shown in (1c). The dorsal stream learns to localize objects by assessing foveated features around random saccadic points and guides the ventral stream in object classification by removing background clutter.

relies on the active interplay of the eyes and two primary pathways: the dorsal (where/how) and ventral (what) streams. According to the two-stream hypothesis, these pathways have distinct specialized functions as depicted in the left column of Figure 1(a). The dorsal stream processes spatial information and guides actions by determining the location and movement of objects, essential for interacting with the environment. Conversely, the ventral stream handles object recognition and form representation, identifying objects based on shape, color, and texture. Additionally, the eyes, equipped with inherent properties such as foveation and saccades, enhance this system by serving as the primary input for the human perception system, particularly in peripheral vision as shown in Figure 1(b). Saccades are rapid eye movements that shift the fovea's focus iteratively to different parts of the visual field, analyzing each area for relevant features. Foveation locks these points to process high-resolution details, enabling object recognition (Eckstein, 2011). Integrating these functionalities into the dual-stream model enhances the system into an active vision model (Curcio et al., 1990; Land & Nilsson, 2012). Thus in an active vision system, the dorsal stream iteratively directs saccades to spatially significant areas, separating the foreground from the background, which are then processed by the ventral stream for detailed recognition.

Humans, existing in a 3D environment, understand objects by identifying and examining the structure from various viewpoints. Complex objects may require multiple views, unlike simpler ones, to provide a robust understanding of their nature. This process is illustrated in the right column of Figure 1(c) with a deep learning-based active vision system. Given an input image with three random saccades, the dorsal stream ($f_D$) analyzes the foveated glimpses for object features, discarding points without salient features. Relevant features prompt foveation expansions to extract the foreground from the background, which is then passed to the ventral stream for object recognition. The dorsal stream learns the concept of the object by viewing it from different saccadic points, determining which views enhance understanding and which to discard. This helps eliminate unnecessary noise, allowing the ventral stream ($f_V$) to focus solely on object features for better recognition, even in the presence of background clutter.

Contrary to active vision systems, current deep learning (DL) methods for image classification process the entire image in one shot, uniformly attending to each pixel. This approach is analogous to the ventral stream analyzing the entire image for object recognition without the guidance of the dorsal stream, leading to the inclusion of both foreground and background features, as shown in Figure 2 (1b). While this passive

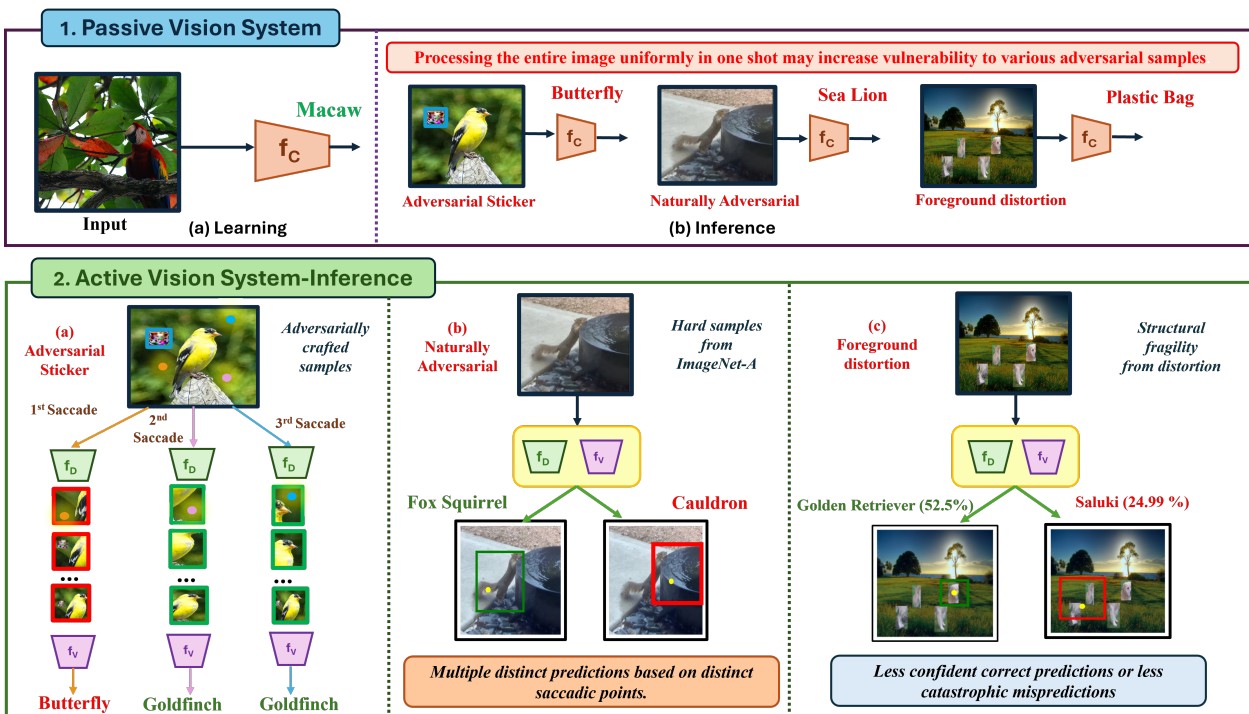

Figure 2: The upper part of the figure illustrates that passive vision systems, mimicking the "what" pathway alone, are more vulnerable to adversarial inputs due to uniform image processing. The bottom part shows the active vision system's enhanced robustness across three categories: adversarially crafted samples (2a), naturally adversarial samples (2b), and foreground distorted images (2c). Active vision systems make multiple correct predictions despite non-uniform noise, identify object-relevant features, and result in less catastrophic mispredictions respectively.

analysis has achieved significant success with benign inputs (Krizhevsky et al., 2012; Ren et al., 2015; He et al., 2017), it struggles with adversarial inputs that include imperceptible noise ignored by human eyes (Szegedy et al., 2014; Goodfellow et al., 2015; Papernot et al., 2016). Depending on their generation process, this imperceptible noise can spread across the image, affecting both the background and foreground, or be tailored to harm only the foreground. Such malicious inputs have been detrimental to various vision-oriented applications (Athalye et al., 2017; Hosseini & Poovendran, 2018; Joshi et al., 2019). Naturally occurring adversarial samples can also reduce classification accuracy to zero, as shown in ImageNet-A (Hendrycks et al., 2021), where objects may be occluded, have a smaller foreground-to-background ratio, or exhibit real-world effects, underscoring the need for a non-passive analysis akin to human vision for complex images.

To address these challenges, various methods have been proposed from both algorithmic and human peripheral vision perspectives. Algorithmic approaches utilize deep learning principles such as Adversarial Training, Randomised Smoothing, Data Augmentation, etc. (Madry et al., 2018; Cohen et al., 2019; Andriushchenko & Flammarion, 2020; Yun et al., 2019; Li & Spratling, 2023; Moosavi-Dezfooli et al., 2018; Yue et al., 2023; Zhang et al., 2018). Inspired by human vision, some research focuses on non-uniform visual processing to mitigate adversarial noise, exploring cortical fixations (Vuyyuru et al., 2020), peripheral blurring (Shah et al., 2023), primal visual cortex processing (Dapello et al., 2020), and fovea-based texture transformation (Gant et al., 2021). These methods, modeled on a single fixation point during training, primarily enhance the ventral stream's passive approach.

Building on the principles of active vision, this work goes beyond the traditional single-point inference approach and advocates that the active vision system, due to the dorsal stream's dynamic focus, that is inherently more robust to various kinds of adversarial inputs. This idea is demonstrated in part (2) of Figure 2 for three different categories of adversarial inputs: crafted adversarial samples, naturally adversarial images, and foreground object distortions. Carefully crafted adversarial noise often has a non-uniform distribution

across the entire input to match imperceptibility (PGD-like attacks) (Szegedy et al., 2014; Wang & He, 2021; Dong et al., 2018) etc. or size (patch-based attacks) (Gao et al., 2020) constraints. [2] By processing inputs through multiple saccades and foveated glimpses, active DL methods utilize the dorsal stream's "where" functionality to make distinct predictions under non-uniform adversarial noise, directing saccades to identify significant areas and discard irrelevant ones. While one saccadic point might lead to a misprediction fooled by the presence of adversarial noise, others often result in robust foveated glimpses, leading to correct predictions as shown in the leftmost column (a). For naturally adversarial samples, which are inherently challenging to classify, the dorsal stream's ability to assess different portions of the image aids in identifying the correct salient features. This capability often leads to correct predictions that would be difficult for a passive, ventral-only approach, as shown in the middle column (b). In the third category, as presented in the rightmost column (c), we further explore the concept of visible adversarial samples by structurally distorting the foreground. This often leads to catastrophic mispredictions for passive methods. Although the active system may not see the complete object, it can still identify meaningful features by separating parts from the background, resulting in less confident correct predictions or less catastrophic mispredictions (e.g., parts of a Golden Retriever being misclassified as a Saluki—both being dog breeds).

This work leverages the frameworks of Active Localization with Foveation and saccades (FALcon) (Ibrayev et al., 2024b) and Glance and Focus Networks (GFNet) (Wang et al., 2020), analyzing them through the lens of an active vision system framework. While FALcon is designed for Weakly-Supervised Object Localization and GFNet for budgeted and anytime inference, both exhibit an underlying active dorsal-ventral structure. However, their inherent adversarial robustness has not been studied prior to this work, making them ideal DL-based active vision candidates for this robustness analysis. For evaluation, we employ a black-box transfer attack setup, where adversarially crafted samples are generated from surrogate models and transferred to the target active and passive vision models. Consistently, for the other two kinds of adversarial inputs, samples are passed for inference. It is important to note that we are not proposing an adversarial defense for a black-box scenario, nor do we claim robustness to query-based black-box attacks.

Hence, our contributions are highlighted as follows:

- We present a **novel analysis** of the **inherent robustness of Active Vision systems** across three different categories of adversarial inputs.

- Our experiments demonstrate that active vision improves the ventral only passive vision's performance by **2-21%** in accuracy against **adversarial crafted inputs** across various state-of-the-art transfer attacks (4.2) on ImageNet (Deng et al., 2009) in a black-box transfer attack setup.

- Through both quantitative and qualitative analyses, we highlight the **salient learning aspects** contributing to the inherent robustness of these methods, including **glimpse-based focused learning at downsampled resolutions** (4.3) and **inference from distinct saccadic points** (4.4).

- We provide similar detailed analysis for the **naturally adversarial ImageNet-A** dataset, demonstrating **1.5-2 times** improvement over the passive ventral method. We provide qualitative results for diverse set of samples within this category.

- We present **qualitative results** that highlight the benefits of **implicit** structured learning in active systems for **handling foreground object distortion**, leading to more human-aligned and interpretable predictions.

## 2  Related Work

**Active Vision methods** The methods discussed here explore the incorporation of active iterative strategies for input processing. RANet (Mnih et al., 2014) incorporates a recurrent attention network to selectively focus on different parts of the input sequence over multiple time steps excelling in sequential tasks. Saccader (Elsayed et al., 2019) emulates saccadic eye movements to iteratively extract features from an image attending

---

[2] Since this noise is imperceptible to the human eye, we use an adversarial sticker for illustration purposes to emphasize the high concentration of adversarial noise in that region.

to finer details while enhancing performance. Glance and Focus Networks (GFNet) (Wang et al., 2020) constrained by computational budget, iteratively processes different glimpses in an image, refining predictions until confidently identifying the object. Foveated Transformer (Jonnalagadda et al., 2022) uses pooling regions and dynamic fixation allocation based on Transformer attention based on past and present fixations for image classification. Recently, FABLE (Ibrayev et al., 2024a) proposed a localization framework that models the ventral stream as a supervised feature extractor and the dorsal stream as a separate model trained via reinforcement learning. Building on this, FALcon (Ibrayev et al., 2024b) enhances the approach by incorporating foveation and saccades, enabling dynamic and active vision for improved object localization and multi-object detection, even when trained on single-object images. In this study, we explore how the iterative interplay of foveation and saccades in a dorsal-ventral system enhances the inherent adversarial robustness of active methods across different adversarial inputs.

**Towards bio-inspired mechanism for robustness** The following methods address the adversarial inputs by functionally treating human eyes as pre-processing/transformation stage. (Luo et al., 2015) demonstrates that applying CNNs to specific foveated regions reduces the impact of adversarial perturbations by leveraging CNNs' robustness to object scale and translation, along with non-linear responses in background regions. On ImageNet, this approach achieves accuracy close to unperturbed levels, even when perturbations are crafted with foveation in mind. R-Warp (Vuyyuru et al., 2020) advocates for biologically inspired mechanisms such as cortical fixations and retinal fixations incorporated in DNNs lead to adversarial robustness for small perturbations. VOneBlock (Dapello et al., 2020), illustrates that incorporating primary visual cortex processing at the forefront of CNNs enhances their resilience against image perturbations. Harrington et al. (Harrington & Deza, 2022) demonstrates that adversarially robust networks behave similarly to texture peripheral vision models, thus promoting the latter's plausibility for adversarial robustness. (Gant et al., 2021) proposed a novel Foveated Texture Transform module in a VGG-11 to enhance adversarial robustness without sacrificing standard accuracy. R-Blur (Shah et al., 2023) simulates peripheral vision using adaptive Gaussian blurring and trains on these transformed input images, leading to improved adversarial robustness. While these methods simulate human peripheral processing, they do not replicate the iterative active learning process found in human vision. We provide a fresh perspective showing how mimicking human-like active vision processing naturally enhances DNN robustness against adversarial inputs.

**Transfer attacks** (Szegedy et al., 2014) introduced the vulnerabilities of neural networks to adversarial samples. (Papernot et al., 2016) introduced a novel approach that leverages substitute models to craft transferable adversarial examples, emphasizing the need for robust defenses against such attacks. (Liu et al., 2017) conducted an extensive investigation into the transferability of adversarial samples on large-scale datasets like ImageNet (Deng et al., 2009). Recently, LGV (Gubri et al., 2022) exploited the weight space geometry of surrogate models to find flatter adversarial samples creating stronger transfer attacks. Token Gradient Regularization (TGR) (Zhang et al., 2023) introduces a method that enhances the transferability of adversarial attacks on Vision Transformers (ViTs) by focusing on Token Gradient Regularization (TGR). This approach manipulates token-level gradients to create perturbations that effectively fool different ViT models, highlighting a significant vulnerability in these architectures. In this study, we examine active vision methods under the lens of adversarial robustness in a black box transfer threat model and show the human-inspired active way of processing inputs in DNNs leads to inherent robustness.

## 3 Active Vision systems

In this section, we provide a focused overview of the inference process and **highlight** key insights into the inherent robustness of two active vision systems: FALcon (Ibrayev et al., 2024b) and Glance and Focus Networks (GFNet) (Wang et al., 2020). These methods simulate foveation by cropping glimpses from the image based on fixation (saccadic) points, without blurring the extracted glimpses. This approach can be interpreted as foveation with an extreme cut-off. For detailed learning processes, readers are directed to supplementary Sections 1.1 (FALcon) and 1.2 (GFNet). For the remainder of this manuscript, we will refer to saccadic points as fixation points.

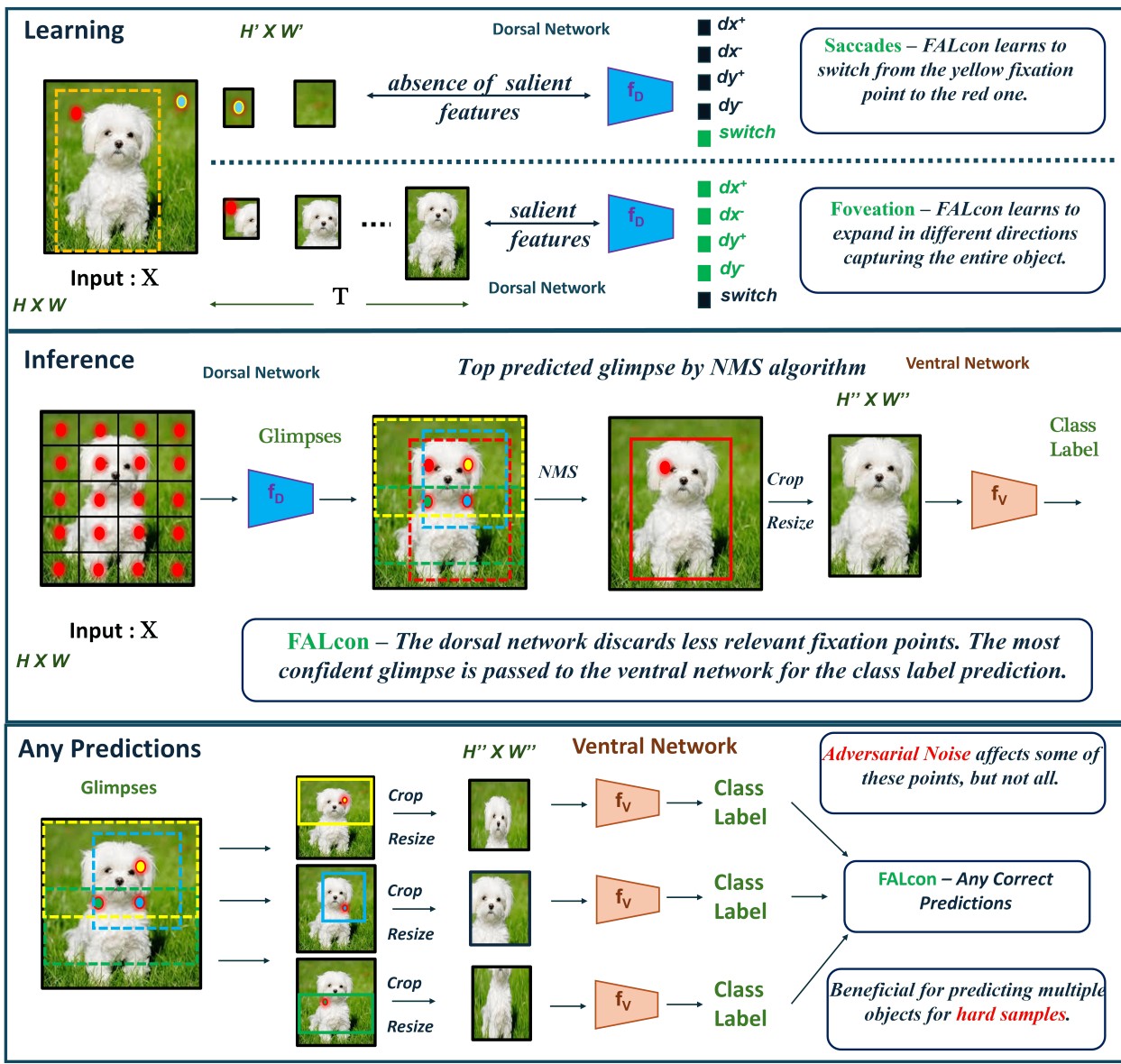

Figure 3: The figure provides a high-level overview of FALcon. During **Learning**, the dorsal network ($f_D$) is trained to predict five distinct actions (four for expansion and one for switching), enabling it to learn the importance of each fixation point illustrated by colored dots. Learning occurs in a downsampled resolution of $(H' \times W')$. During **Inference**, $f_D$ starts from each pre-defined multiple fixation point (20 red dots). If salient object features are present, $f_D$ performs the learned expansions to capture the object (4 colored dashed boxes, colored dots). The most confident final foveated glimpse (red solid box) is cropped ($H'' \times W''$) and presented to the ventral ($f_V$) for **Top** prediction. The system can produce distinct predictions referred to as **Any**, which are beneficial for handling adversarial samples.

## 3.1 FALcon

**Active Vision structure** Both the dorsal $f_D$ and ventral $f_V$ streams are represneted by deep convolutional neural networks. During training, only $f_D$ is trained to emulate the saccadic and foveated functions. For $f_V$, any pre-trained network can be selected.

**Inference** During inference, the input image $X$ is divided into grid cells, as illustrated in the first image at the bottom part of Figure 3. Each grid cell is considered as an initial fixation point (red dot). The

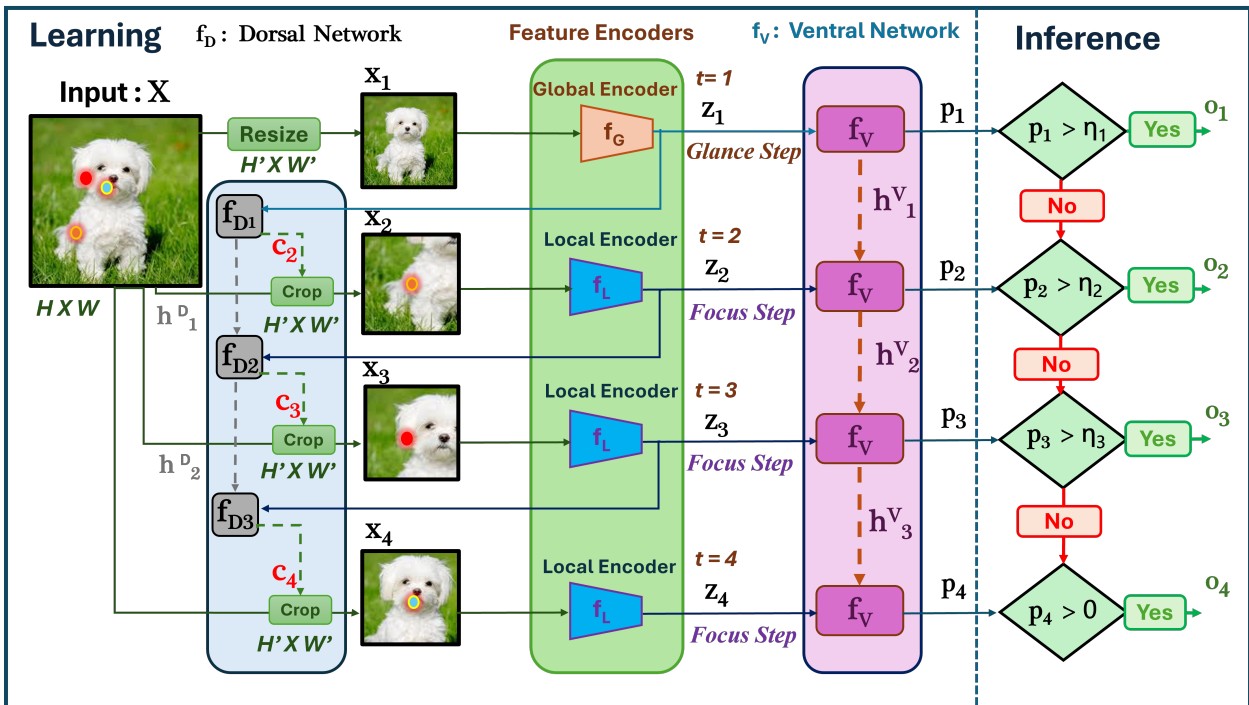

Figure 4: (**Learning & Inference**) The figure provides an overview of GFNet's operation. It begins by downsampling the input image to a lower resolution for rapid prediction ($p_1$), termed Glance at $t = 1$. If the network lacks confidence ($p_1 < \eta_1$), it enters subsequent Focus steps until certainty is attained or till ($t = 4$). Each focus step analyzes a patch ($H' \times W'$) cropped from the original input ($H \times W$) centered around ($c_t$) illustrated by colored dots. These co-ordinates are determined by the dorsal network $f_D$. The process is depicted for a sequence length of 4.

dorsal $f_D$ initiates from each point with a pre-defined glimpse size to inspect salient object features. In the absence of salient features, the dorsal deems there is no potential object and hence **switches** to another fixation point. If the dorsal encounters any relevant features and hence assumes the presence of a potential object, it expands from the initial glimpse dimensions in four independent directions to generate a sequence of foveated glimpses. Based on the expansion operations, each foveated glimpse is cropped from the input image at a **downsampled resolution of** $H' \times W'$. The sequence of expanding for a potential fixation point continues until the final foveated glimpse **captures the entire object** learning the structure implicitly. This process iterates for all potential points that could lead to various final foveated glimpses for the same object. This is illustrated by the colored dots and dashed colored boxes in the second image. For object localization, these final foveated glimpses undergo non-maximum suppression to yield the most confident prediction indicated by the solid red bounding box in the third image. The region corresponding to the most confident final foveated glimpse is cropped **at standard dimensions** $H'' \times W''$, and then presented to the ventral network $f_V$ for class prediction. In our evaluation, we refer this final class prediction label as **Top**. The remaining potential fixation points, except the most confident one, serve as a map to understand the effect of non-uniform adversarial noise injected into images. This is illustrated at the bottom part of the figure under **Any Predictions**. In our experiments, we visually demonstrate how the noise affects some of these fixation points (4.4), but not all, resulting in enhanced robustness, which we quantitatively refer to as **Any**.

### 3.2 Glance and Focus Networks

**Active Vision structure** GFNet potrays a slightly more complex framework as DL based active vision system. Both the dorsal $f_D$ and ventral $f_V$ streams are represented as deep recurrent neural networks to

aggregate information from previous steps. For processing image features, both streams employ convolutional deep neural network based feature encoder backbones $f_{G/L}$. In the training phase, all networks are trained.

**Inference** GFNet performs inference in two distinct steps – a glance step and subsequent multiple focus steps (Figure 4). In the glance step, the full-resolution image ($H \times W$) is first **downsampled to a much lower resolution** ($H' \times W'$). It is then passed through the global encoder $f_G$ and ventral network $f_V$ pathway to make a swift prediction based on the global features. If the confidence $p_t$ exceeds the threshold $\eta_t$, where $\eta$ is a pre-defined threshold (Huang et al., 2018; Yang et al., 2020), the process halts. Concurrently, the dorsal network $f_D$ evaluates these features to predict the fixation point for the subsequent focus step. Each foveated glimpse is generated based on the most salient features of the object centered around the fixation point, denoted by the orange dot in the second glimpse. This $H' \times W'$ glimpse is cropped from the image and inputted into the local encoder $f_L$ ventral network $f_V$ pathway for prediction in the second step. Simultaneously coordinates for the next focus step are produced by the dorsal network. The iterative process persists until the network gains sufficient confidence in its prediction, or reaches the end of the sequential process $t = T$. To understand the robustness aspect through inference on transferred adversarial samples, we keep the early termination inactive, allowing inference to continue until $t = T$. This allows the ventral network to process glimpses cropped around distinct fixation points generated by the dorsal network, at each step assessing the input in the presence of adversarial noise and background clutter. Both $f_G$ and $f_L$ process **low dimensional inputs of** $H' \times W'$ with the former fine-tuned on these dimensional inputs for global step predictions. Exploiting this in our experiments, we demonstrate how learning in a downsampled resolution contributes to the robustness properties of such systems 4.3.

## 4    Adversarial crafted images

**Adversarial transfer attack setup** The section aims to illustrate that active vision networks GFNet (Wang et al., 2020) and FALcon (Ibrayev et al., 2024b) exhibit higher levels of robustness against transferred adversarial images than base passive classifiers (He et al., 2015). We follow the protocol for a black box transfer attack threat model as outlined in (Liu et al., 2017; Mahmood et al., 2021). Following this protocol, we define non-targeted transferability. Given a surrogate Classifier $S_i$, we generate an adversarial sample for an image/label pair $(x, y)$ which is denoted as $x_{adv}$. This is with respect to the surrogate Classifier $S_i$ and attack pair $A_{S_i}$. The adversarial sample, $x_{adv}$, is said to transfer to another target Classifier $T_i$ if the adversarial sample is mispredicted. This is formalized as the following:

$$x_{adv} = A_{S_i}(x, y) \mid S_i(x_{adv}) \neq y; \qquad T_i(x_{adv}) \neq y \tag{1}$$

**Metrics** We measure the non-targeted transferability by computing the percentage of adversarial examples generated using model $S_i$, but still correctly classified by the model $T_i$ (not transferred). We refer to this percentage as accuracy. A higher accuracy means less susceptibility to transferred adversarial samples and hence higher robustness under this setup. For a test set with $N$ samples, the accuracy is defined as:

$$Acc_{s \rightarrow t} = \frac{1}{N} \sum_{j=1}^{N} \mathbb{1}\{T_i(x_{\text{adv}_j}) = y_j\} \tag{2}$$

**Remark** In this study, we focus solely on empirically showcasing the inherent robustness of active vision methods. We do not propose any adversarial defense for a black-box attack scenario or analyze the transferability trends between surrogates and target samples. Therefore, we opt for standard accuracy ($Acc_{s \rightarrow t}$) under transfer, where we denote $s \rightarrow t$ as surrogate to target.

Section (4.2) empirically demonstrates this via quantitative results. Sections (4.3) and (4.4) then focus on explaining the salient features that provides this inherent robustness, by analysing the internal mechanics of GFNet and FALcon, respectively, in the presence of transferred adversarial inputs.

Table 1: Inherent Robustness of Active Vision Methods

| **Clean** | | ResNet50 | CutMix | FALcon-Top | **Adv-T★** | **GFNet** |
|---|---|---|---|---|---|---|
| | | 76.15 | 78.60 | 72.97 | **47.91** | **75.88** |
| Surrogate | Target | Attack | | | | |
| | | PGD | MIM | VMI-FGSM | P-IFGSM | TI-FGSM |
| ResNet34 | ResNet50 | 31.46 | 20.20 | 11.61 | 31.62 | 28.08 |
| | **FALcon-Top** | **49.83** | **37.01** | **29.40** | **40.46** | **35.20** |
| | **FALcon-Any** | **53.76** | **41.28** | **33.74** | **44.72** | **39.08** |
| | CutMix | 43.47 | 30.92 | 21.32 | 41.34 | 38.78 |
| | **FALcon-CutMix-Top** | **52.48** | **39.88** | **31.97** | **44.21** | **42.04** |
| | **FALcon-CutMix-Any** | **55.86** | **43.95** | **36.25** | **48.07** | **45.96** |
| | Adv-T★ | 47.63 | 47.37 | 47.16 | 46.85 | 46.43 |
| | **GFNet** | **57.82** | **48.60** | **41.17** | **46.93** | **41.42** |
| ResNet50 | ResNet50 | 0.00 | 0.00 | 0.00 | 0.00 | 0.00 |
| | **FALcon-Top** | **31.73** | **19.96** | **12.07** | **27.37** | **14.63** |
| | **FALcon-Any** | **37.54** | **25.60** | **16.64** | **32.37** | **19.14** |
| | **GFNet** | **51.85** | **42.16** | **32.33** | **43.33** | **34.30** |

| Attack | Source | GFNet | FALcon-Top | FALcon-Any | Source | FALcon-Top | FALcon-Any |
|---|---|---|---|---|---|---|---|
| **AutoPGD** | **ResNet50** | 56.54 | 31.00 | 37.80 | **VGG16** | 52.98 | 57.11 |

Table 2: AutoPGD attacks transferred from ventral and dorsal architectures of Active Vision System

## 4.1 Implementation details

We perform our extensive robustness analysis on Imagenet (Deng et al., 2009), a standard benchmark for image classification. We utilize ImageNet pre-trained weights for GFNet and FALcon without any additional fine-tuning. Following the active vision structures highlighted in Section 3, we employ GFNets with ResNet50 as both global $f_G$ and local $f_L$ encoders. These encoders provide relevant image features to the dorsal and ventral streams which are recurrent neural networks, as illustrated in Figure 4. Both encoders are trained on downsampled resolution images of (96, 96) pixels. For FALcon we employ VGG16 (Simonyan & Zisserman, 2015) as the dorsal $f_D$ stream, and ResNet50 as ventral $f_V$. Please note that, unlike GFNet, the $f_V$ of FALcon is not involved during training. Instead, only the dorsal is trained on the downsampled images, while a pre-trained ResNet50 is employed as $f_V$ during inference on image resolutions of (224,224) as indicated by $H" \times W"$ in Figure 3. This approach provides the flexibility to select various ventral streams and demonstrate how the FALcon framework can enhance the performance of the underlying passive ventral networks. We utilize TORCHATTACKS (Kim, 2020), an integrated library for generating adversarial attacks (Ravikumar et al., 2022) with PYTORCH, to generate adversarial samples.

## 4.2 Inherent robustness in the Black-box transfer attack setup

In this section, we demonstrate the superior performance of active vision systems (e.g., FALcon and GFNet) over passive ventral ones (e.g., supervised ResNet) in a black-box transfer attack setup. Adversarial samples generated from surrogate models are transferred to the unknown target models. For GFNet, we use the output from the final prediction step as described in Section 3.2. For FALcon, we evaluate two types of predictions: **Top**, where the most confident prediction is matched with the ground truth, and **Any**, where any correct prediction from multiple outputs is considered. This is enabled by inference from multiple

distinct fixation points. The **Any** prediction strategy highlights the full potential of active systems, detailed in Section 3.1.

**Iterative attacks** We generate adversarial samples from surrogate classifiers $S_i$ using iterative adversarial attacks such as PGD (Madry et al., 2018), MIM (Dong et al., 2018), VMI-FGSM (Wang & He, 2021), Patchwise-IFGSM (Gao et al., 2020) and TI-FGSM (Dong et al., 2019). In the first setup, ResNet34 is used for $S_i$, and in the second setup, ResNet50 is used matching the ventral stream in the active vision networks. For GFNet, ResNet50-based samples attack both dorsal and ventral streams simultaneously. FALcon, however, offers more control over which stream is targeted, as detailed in Auto-PGD 2. We can also substitute FALcon's ventral stream; for instance, **FALcon-CutMix-Any** uses the default VGG16 $f_D$ stream but replaces the $f_V$ with a ResNet50 trained using CutMix loss (Yun et al., 2019). The performance is then evaluated based on any correct prediction matched with the ground truth. We conduct $L_\infty$ attacks with 10 iterative steps, $\alpha = 2/255$, and $\epsilon = 8/255$ for all six iterative attacks including Auto-PGD (Croce & Hein, 2020). Adversarial samples are generated using the entire 50,000-sample ImageNet test set. The corresponding clean accuracy is presented at the top of Table 1.

**Quantitative analysis** Table (1) demonstrate that active vision systems consistently improve upon the underlying passive approach across all surrogate architectures and attacks. For instance, FALcon with a Supervised-ResNet50 and a CutMix-ResNet50 ventral stream shows steady performance improvements of approximately **11%-23%** and **7%-15%** in accuracy, respectively, over the corresponding passive ventral backbones, proportional to the attack strengths. Specifically, for a supervised-ResNet50, FALcon-Top improves performance by nearly **18%** for PGD, while for CutMix, FALcon-Any achieves close to a **15%** improvement for VMI. CutMix (Yun et al., 2019) has robustness properties stemming from its strong regularized feature representations as indicated by the higher baselines than supervised-ResNet50. Additionally, we consider an adversarially trained ResNet50 Madry et al. (2018), which serves as an Oracle method denoted as Adv-T$\star$. Trained specifically for adversarial defense, this method offers the best-case performance on transferred samples on average. In the second setup, we notice that FALcon and GFNet provide an additional shield, even when the attack is generated using the ventral backbone and shared feature encoder respectively. While GFNet employs a more complex framework with recurrent dorsal and ventral streams sharing a convolutional backbone, FALcon-Top offers a clear measure of quantitative improvement due to its active processing mechanism. As shown in Table 2, the non-zero results demonstrate robustness benefits even when generating Auto-PGD-based adversarial samples using each active vision system's crucial networks. This table indicates that generating samples based on FALcon's ventral stream is more effective than using its dorsal stream. This is pictorially explained in Section A.1.3.

**Transfer attacks with Large Geometric Vicinity (LGV)** The plot on Figure 5 presents results based on a geometric space attack (Gubri et al., 2022). The intuition behind this attack is provided in in Appendix A.1.4. We follow a similar experimental setup as outlined in the paper (Gubri et al., 2022), combining LGV with PGD and BIM (Kurakin et al., 2018) on 1000 randomly sampled images from the ImageNet validation set. We report accuracy ($Acc_{s \rightarrow t}$), and the results indicate a consistent trend similar to the iterative attacks for supervised-ResNet50 and CutMix-ResNet50. For instance, as depicted in the plot, FALcon-CutMix-Any improves upon Top by 3-4%, which in turn improves upon the baseline by 24-28% for BIM (orange) and PGD (blue), respectively. Conversely, when FALcon is paired with an adversarially trained ResNet50 (Madry et al., 2018), we observe close to a 2% improvement on clean samples (notably low accuracy for adversarially trained models) but no significant improvement for adversarial samples. This is expected, as networks already trained on worst-case perturbed samples benefit less from active predictions based on distinct fixation points.

**Transfer attacks with Token-Gradient Regularization (TGR) Setup** We follow the experimental setup outlined in the original paper (Zhang et al., 2023) and present results on a test set of 1,000 randomly selected images from the ImageNet validation set (Deng et al., 2009). The equation and the intuition behind this attack are provided in Equation (3) in Appendix A.1.5. The TGR transferable attack plot in Figure 5 shows the accuracy ($Acc_{s \rightarrow t}$) of different target networks on adversarial samples transferred from various surrogate architectures, including Vision Transformers (ViT-B/16) (Dosovitskiy et al., 2021), and their variants PiT-B, and CaiT-S/24 (Touvron et al., 2021; Heo et al., 2021). In addition to the baseline methods previously studied, we employ several notable vision transformer architectures for image classification, such

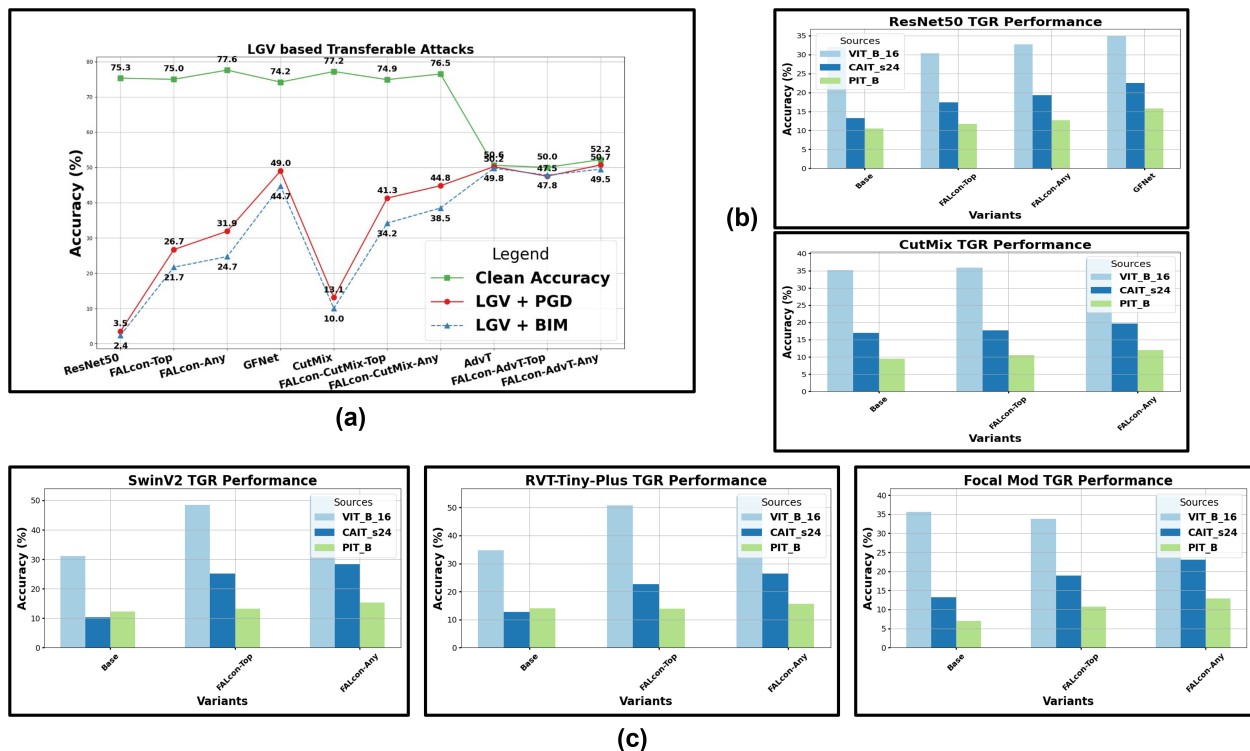

Figure 5: (a) The scatter-line plot on the left shows accuracy improvements in LGV-based transfer attacks using a ResNet50 surrogate, with active models significantly outperforming passive ones, except for the adversarially trained ventral. The bar plots on the right (b) and bottom (c) illustrate the effect of Token Gradient Regularization on various ventral streams. While CNN-based streams show modest gains (b), significant improvements are seen in CNN (dorsal)-ViT (ventral) configurations (c).

as Swin V2 Transformer (Liu et al., 2022), Focal Modulation Networks (Yang et al., 2022), and Robust Vision Transformer (Mao et al., 2021), as ventral methods combined with FALcon's default VGG16 dorsal stream. The vision transformers examined are their corresponding tiny versions, with a parameter count close to 25 million parameters, comparable to the CNN baselines studied.

**Results** For FALcon-SwinV2-Tiny and FALcon-RVT-plus-Tiny, active systems demonstrate significant improvements on adversarial samples transferred from ViT-B/16 and CaiT-S/24, with gains of **16%** and **14%**, respectively. For Focal Modulation networks, we see the active vision method shows significant gains mostly for CaiT-S/24 based samples. This highlights the effectiveness of a CNN-Transformer-based dorsal-ventral active system in handling such token based transferable samples. For PiT-B-based adversarial samples, improvements across all architectures range between **2-6%**. Although CNN-based baselines also show steady gains, they are less pronounced compared to those with transformer-based ventral backbones.

**Key takeaway** Across various transferred adversarial samples, an active vision system consistently improves upon the underlying passive ventral stream for both CNN and Transformer based architectures. However, this approach shows limitations when applied to adversarially trained ventral streams (FALcon-AdvT) as indicated by Figure 5 (a), highlighting the need for further research into adversarially trained active dorsal networks. In Appendix A.1.2, we provide an additional discussion on adversarially trained models.

In the following sections, we present the primary factors contributing to this enhanced robustness: processing inputs in a down-sampled resolution (Section 4.3) and performing inference from different fixation points (Section 4.4).

Table 3: Effect of glimpse-based learning on downsampled resolutions.

| Setting | Surrogate | Target | PGD | | MIFGSM | |
|---|---|---|---|---|---|---|
| | | | Resolution | | | |
| | | | (96,96) | (128, 128) | (96,96) | (128,128) |
| 1 | **Clean** | ResNet50 | 52.42 | 64.42 | 52.42 | 64.42 |
| | | GFNet | **75.88** | **76.70** | **75.88** | **76.70** |
| 2 | ResNet34 | ResNet50 | 46.03 | 54.10 | 41.05 | **45.73** |
| | | GFNet | **57.82** | **55.46** | **48.60** | 44.72 |
| | ResNet50 | ResNet50 | 46.00 | **51.41** | 41.07 | **42.31** |
| | | GFNet | **51.85** | 45.98 | **42.16** | 34.76 |
| 3 | ResNet34 | ResNet50 | 13.24 | 19.64 | 8.17 | 12.33 |
| | | GFNet | **34.40** | **36.24** | **24.30** | **24.95** |
| | ResNet50 | ResNet50 | 0.30 | 0.13 | 0.35 | 0.16 |
| | | GFNet | **17.96** | **12.12** | **10.63** | **6.80** |

### 4.3 Effects of glimpse-based downsampling (case study: GFNet)

In this section, we use GFNet to explore how learning image representations based on glimpses at a down-sampled resolution contributes to the inherent robustness. Downsampling inherently causes reduction in features. Adversarial imperceptible noise is crafted based on the image in its original resolution (e.g. $224 \times 224$). Hence downsampling the image, distorts the noise along with it, thereby reducing its overall impact on predictions. As a result, it is probable to think that an inherent robustness offered by models processing an image via downsampled resolution stems from the distortions on the non-uniform adversarial noise. To analyse this factor we organize experiments in this section into 3 settings:

- **Setting** 1 *Effect of processing downsampled clean images* - Images from the test set are used for evaluation without any adversarial attack. The images are downsampled to $(96, 96)$ and $(128, 128)$ and inference is performed.

- **Setting** 2 *Reduction of efficacy of adversarial noise post downsampling* - Adversarial images are first generated from full resolution images of $(224, 224)$ and then downsampled to $(96, 96)$ and $(128, 128)$, separately, for inference. for an active vision method such as the GFNet, this is an inherent step of their learning and inference pipeline. However, for passive vision methods, we resize the adversarial inputs to match the resolutions separately.

- **Setting** 3 *Generating adversarial attacks on downsampled images* - The images are downsampled to lower resolutions first and then adversarial inputs are generated. These adversarial downsampled inputs are then passed for inference on both passive and active target models. Since downsampling is performed first, the adversarial effect is not downgraded.

For the passive target baseline, we use a ResNet50 pre-trained on ImageNet at resolutions of $224 \times 224$. For GFNets, we infer with two separate models trained on $96 \times 96$ and $128 \times 128$ resolutions. Notably, we maintain consistency by evaluating GFNets on images of matching resolutions. To illustrate downsampling effects, passive baselines are tested on downsampled images of 96 and 128 resolutions (see Table 3). For simplicity, we further refer to GFNets trained on $96 \times 96$ dimensions as "GFNet-96".

**Results** Table 3 presents quantitative results, focusing on $Acc_{s \to t}$. The best performing models are high-lighted in bold. For Setting 1, a passive model trained on a higher resolution suffers a drop in performance when evaluated at downsampled input, unlike GFNets trained for downsampled resolutions. Setting 2 shows that simply downsampling adversarial images to lower resolutions is beneficial. This indicates along with the image resolution, the imperceptible adversarial noise also probably gets downsampled thereby reducing its effect on model predictions even when $T_i$ is same as $S_i$. Furthermore, under this setting, GFNet-96 ex-hibits greater inherent robustness than GFNet-128 when compared to their corresponding passive baselines.

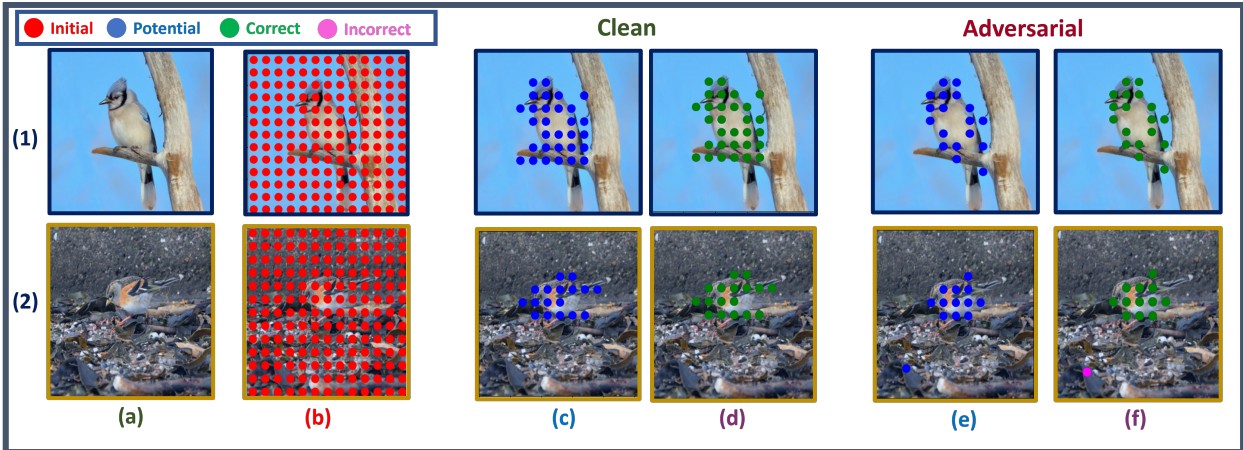

Figure 6: Figure illustrates Initial Fixation Point Maps (**IFPM**) to show the efficacy of performing inference from multiple fixation points. An **IFPM** is a visual representation that depicts the spatial locations of the initial starting positions of FALcon. (**b**) illustrates all initial fixations points via gridding for both clean and adversarial inputs. (**c & d**) show the potential **and** evaluated initial fixation points for a clean sample. Similarly, (**e & f**) show the same for an adversarial sample. An evaluated **IFPM** can consist of both correct and incorrect points as denoted by 2f. Adversarial noise spreads non-uniformly across an image and affects different initial points differently. This is indicated by the reduced number of potential (**c to e**) and correct points (**d to f**) from a clean **to** an adversarial sample. Still, the presence of a positive number of correct points (**f**) underscores the inherent robustness of an active method.

For Setting 3, it is evident that all target models suffer a drop in performances indicating that generating adversarial inputs at resolutions corresponding to target models leads to more potent attacks. Remarkably, GFNet-96 and GFNet-128 demonstrate performance improvements close to 3× and 2×, respectively for ResNet34 as $S_i$, compared to their corresponding passive baselines on downsampled adversarial samples. This further emphasizes the effectiveness of learning in a downsampled regime even under the presence of adversarial attacks.

### 4.4   Effect of distinct fixation points (case study: FALcon)

In this section, we use FALcon to demonstrate the effect of processing an image from distinct fixation points on the robustness of active vision methods. The capability of FALcon to consider various fixation points is used to extract interpretable visualization results. Moreover, since the ventral model is not fine-tuned during training, it allows for a fair comparison with passive baseline network.

### 4.4.1   Initial Fixation Point Map

In order to understand the impact of adversarial noise on regions of the image that influence model predictions, we define an Initial Fixation Point Map (IFPM). IFPM displays the distribution of initial fixation points based on how each of them affects the decision-making of FALcon throughout the inference process. Figure 6 shows IFPMs generated for both clean and adversarial images. As described in Section 3.1, FALcon processes every input from multiple initial fixation points. Red dots indicate all initial fixation points, equally distributed over the image dimensions. Each point is then presented to the dorsal, which retains only those, indicated by blue dots, that potentially resulted in the capture of an object through the series of expanding foveated glimpses.

The ventral processes the final foveated glimpses that resulted from potential points to determine the class label of an object. As a result, various fixation points result in FALcon making correct or incorrect output predictions, indicated by green and magenta dots, respectively. By obtaining IFPM for clean and adversarial versions of the same image, we illustrate how the adversarial noise impacts FALcon in terms of its capacity to make correct predictions from various fixation points.

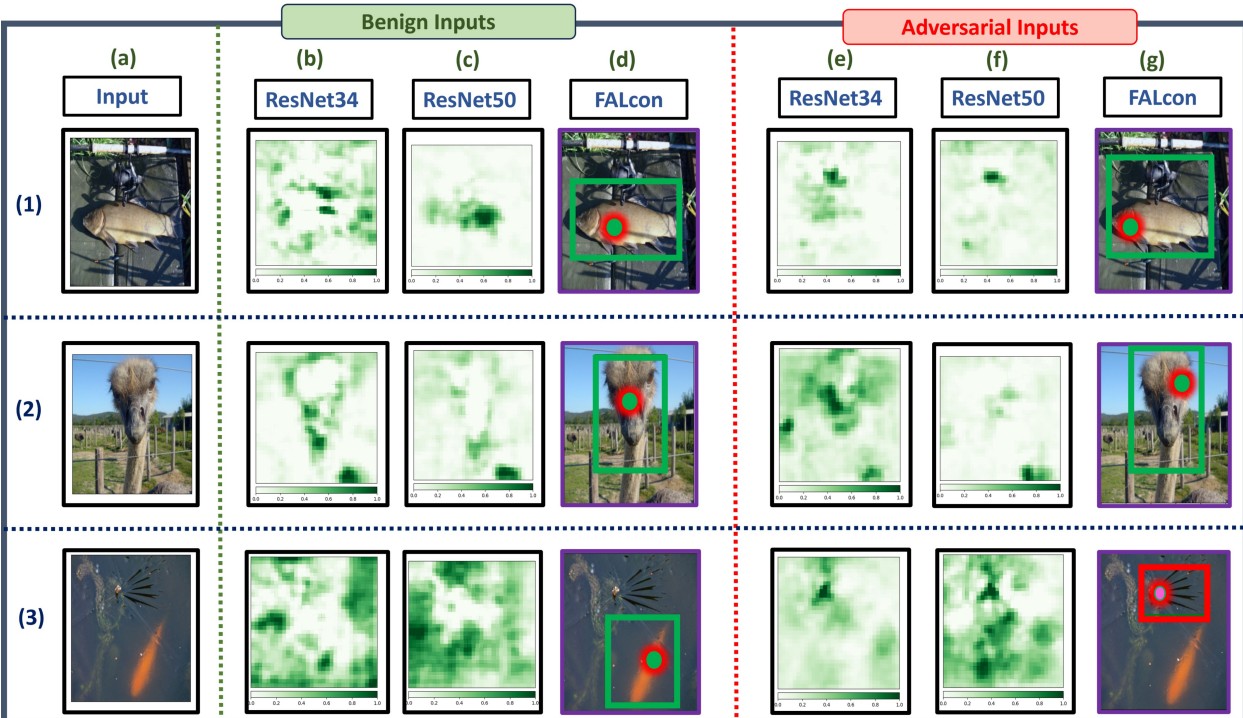

Figure 8: The figure illustrates vulnerabilities of passive methods to adversarial samples. For passive methods **(b,c,e,f)**, occlusion maps are provided, highlighting areas responsible for model predictions. For FALcon **(d,g)**, the final foveated glimpses along with the corresponding initial fixation points are presented. Even though adversarial noise affects an image non-uniformly, passive methods struggle to evade the noise as they process the entire image with equal importance. Contrary, the final foveated glimpse highlights the effect of adversarial noise, guided by the corresponding fixation point.

**Results** IFPMs illustrated in Figure 6 show that despite the addition of adversarial noise, multiple initial fixation points result in correct final predictions (d & f). IFPM clearly indicates the reduced number of potential and correct points for an adversarial sample compared to the corresponding clean sample (c to e) and (d to f). Due to its non-uniformity and imperceptible criteria, the adversarial noise does not affect each point equally. Hence, multiple fixation points lead to correct class predictions. This visually explains the reason for the improved performance of an active method over a passive one, supporting the quantitative results presented in the previous sections. Although noise affects the method, its inherent processing from multiple fixations makes it less susceptible (f). In the second sample (2f), we can notice of a magenta fixation point far away from the object. This is not present for the clean sample and is a false positive due to the addition of the noise. Yet, around the object, we can see multiple green points indicating correct prediction. This validates the hypothesis presented in the first column in Figure (2) (a).

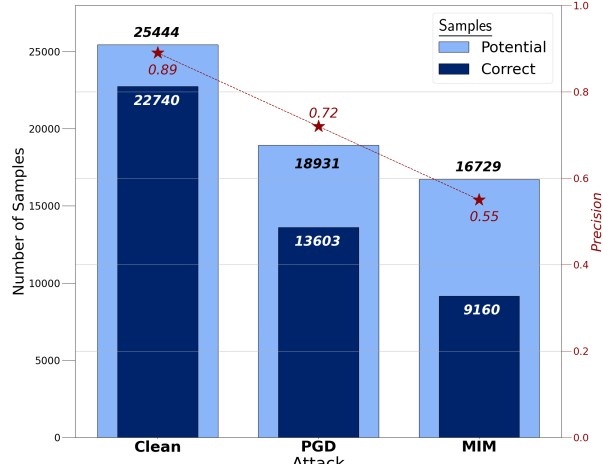

Figure 7: Precision of predictions: The number of potential and correct prediction points decreases as attack strength increases, reflecting the quantitative results presented in Table 1.

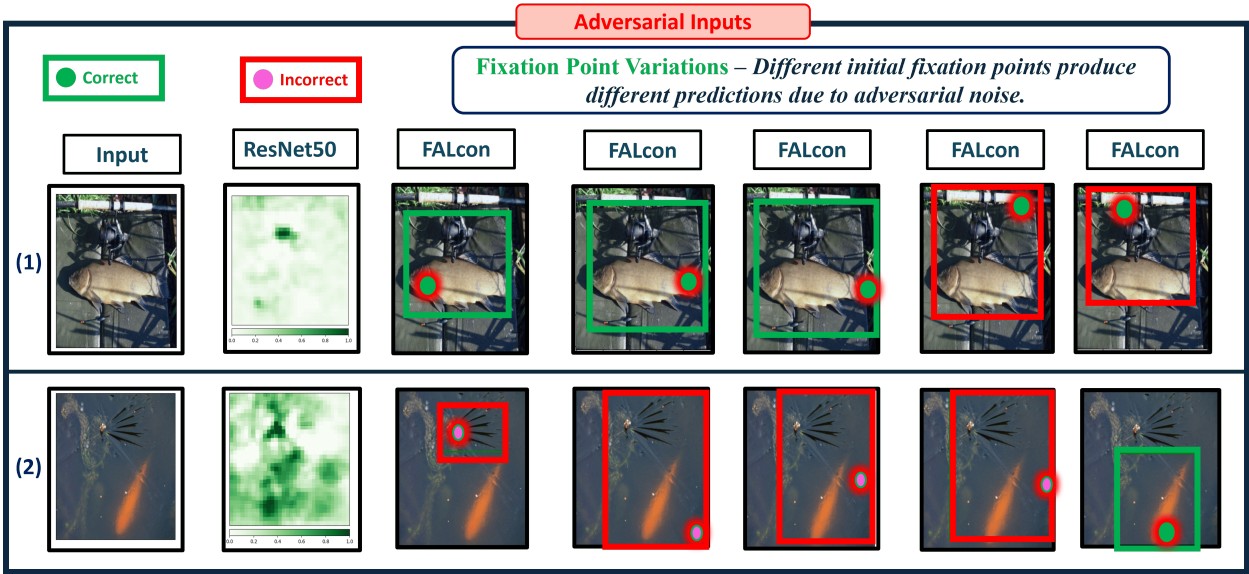

Figure 9: The figure illustrates how the final foveated glimpse changes with different fixation points in the presence of adversarial noise. In sample (1), most predictions are correct (indicated by the solid green box), whereas in sample (2), the opposite is observed (indicated by the solid red box). This variation occurs due to the relative position of the initial fixation point and the regions most influenced by adversarial noise, as shown in the occlusion maps for ResNet50. Final foveated glimpses with a larger background-to-foreground ratio, especially when capturing adversarial noise, typically result in mispredictions by the ventral stream.

**Precision of predictions** In addition to accuracy ($Acc_{s \rightarrow t}$), the ratio of *correct/potential* points serves as another metric for evaluating enhanced robustness. Here, potential points are defined as the sum of true positives (green) and false positives (magenta) as illustrated by an Initial Fixation Point Map (IFPM). For this analysis, we generated attacks on 1,000 images using ResNet34 as the surrogate model. As shown in Figure 7, the high ratio observed for clean samples decreases for adversarial samples, depending on the strength of the attack. Despite this, the persistence of a high number of true positives quantitatively justifies FALcon's improved performance on adversarial samples. This trend is consistent with the results presented in Table 1 for iterative attacks.

### 4.4.2 Explaining adversarial vulnerability of passive methods

**Setup** As mentioned earlier, the probable cause of adversarial vulnerability of the passive vision methods is that they process an input in one-shot with uniform resolution, where every input pixel is treated with the same importance. This is visually demonstrated in this section via occlusion maps. Figure 8 illustrates occlusion maps for passive methods (b,c,e,f) and the final foveated glimpse for FALcon (d,g). An occlusion map is a visual heatmap indicating key regions of an image when occluded, affect the model performance. The darker the region, the higher contribution it has on the final prediction. Occlusion maps are generated based on prediction labels. We first generate the adversarial sample and then generate the occlusion map based on the predicted adversarial label. We use ResNet34 as the surrogate model and PGD as the candidate adversarial attack. The occlusion maps under ResNet50 (f) are based on the transferred adversarial samples from ResNet34. Similarly, for FALcon, we present the final foveated glimpse and the initial fixation point based on the transferred adversarial samples (g). Green solid boxes refer to correct predictions.

**Results** For the clean samples, FALcon correctly predicts all three instances (d). The dark region (1c) aligns with the body of the correct class (tench), resulting in a correct classification for ResNet50. But as indicated in (1f), the dark region shifts and does not align with the body of the object after the injection of adversarial noise leading to an incorrect prediction. For FALcon, although the final foveated glimpse captures the corresponding dark region, the initial fixation point is directed towards the head of the object (Figure 1g). This indicates that FALcon was initially guided by more salient features of the object before encountering the probable adversarial patch later. Additionally, downsampling likely mitigates the impact

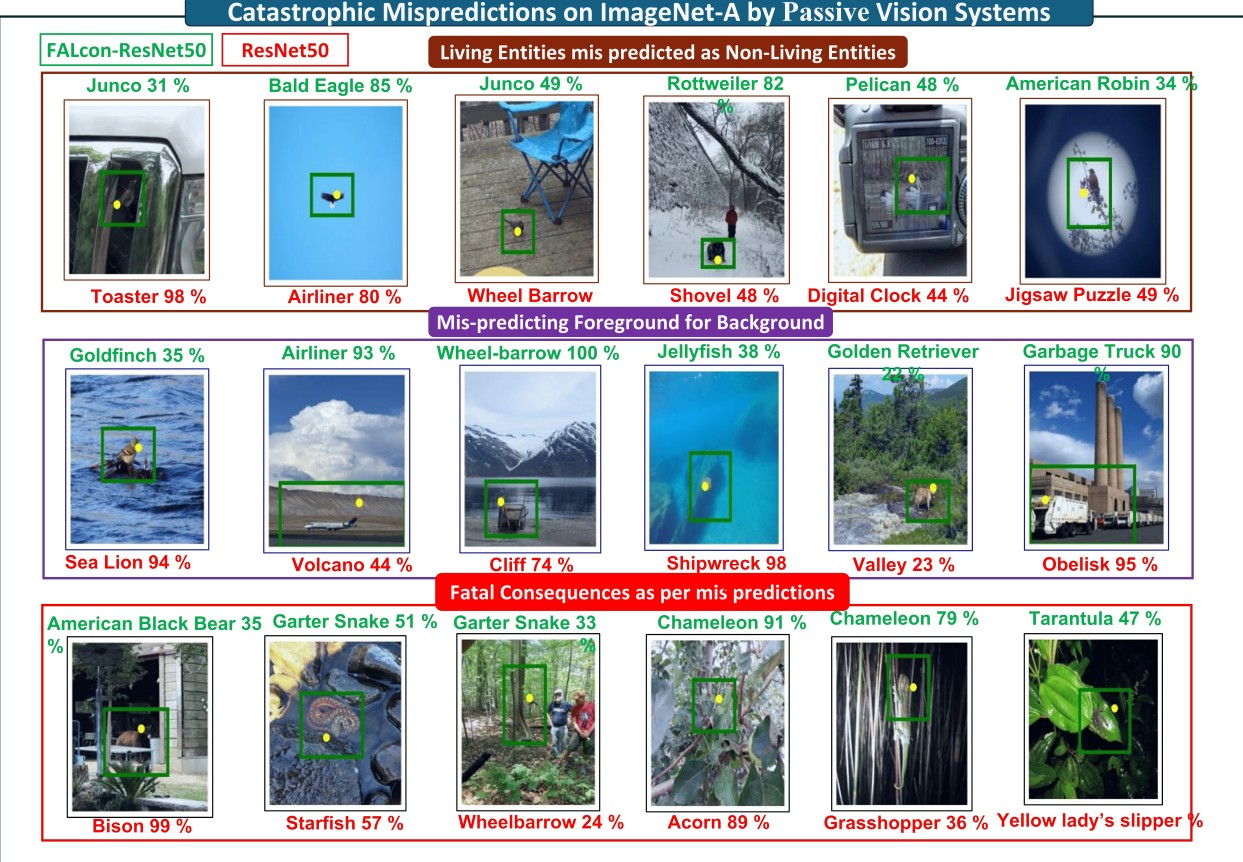

Figure 10: Figure illustrates three categories of catastrophic predictions made by passive vision systems on ImageNet-A. The correct prediction label with the final foveated glimpse is shown in green, while misprediction labels are presented at the bottom of each image. In the first row, living entities are misclassified as non-living entities. The second row shows passive systems getting confused by backgrounds, and the third row highlights fatal consequences caused by catastrophic mispredictions.

of the adversarial patch on the dorsal stream. The final foveated glimpse captures the entire object, with the foreground occupying the majority of the glimpse, thereby minimizing the effect of the adversarial spot in the background and resulting in a confident, correct prediction for the ventral stream. For (3g), FALcon similarly to ResNet50 focuses on the dark region (3f) makes an incorrect prediction highlighted by the red box not capturing the goldfish at all. This suggests that although less vulnerable, there is still room for further improvement for these active vision methods.

**Fixation point variations** The FALcon framework allows us to select initial fixation points to begin the inference process. In the first row, second column of Figure 9, the occlusion map reveals a dark spot representing adversarial noise shown earlier. Since this noise is contained in a small portion of the background, multiple fixation points on the body of the tench lead to correct predictions. However, in the second image, the uneven distribution of noise results in multiple incorrect final foveated predictions by the ventral stream. The glimpses suggest that the dorsal stream is misled, capturing both foreground and background, and consequently fails to effectively guide the ventral stream.

## 5 Natural adversarial images

**Active Vision for hard samples** In this section, we provide insights into the enhanced adversarial robustness of active vision systems when applied to naturally occurring images from the benchmark ImageNet-A dataset (Hendrycks et al., 2021). Unlike adversarially crafted samples, this dataset contains naturally occur-

ring, unmodified images that are difficult for standard passive classifiers to classify correctly, often reducing their accuracy on this challenging set to near zero. The high variability in these images arises from factors such as occlusions, unusual poses, backgrounds that confuse classifiers, or complex scenes with multiple objects where the primary object is not centrally located or is partially obscured. When humans encounter complex scenes, we assess the scene multiple times, iterating from various viewpoints to evaluate each detail before predicting its contents. This iterative and detailed evaluation process, emulated by active vision systems, allows for better handling of the complexity and variability in ImageNet-A, a principle not shared by current passive classification networks. Hence, we empirically show the need for active vision in handling hard samples.

**Catastrophic mispredictions by passive classifiers** This is qualitatively explained in Figure 10, which shows the catastrophic nature of mispredictions made by passive classifiers on this set as opposed to FALcon. We visually categorize the nature of these mispredictions in three separate rows. The final foveated glimpse includes the initial fixation point prediction on the image, the correct active prediction in green (FALcon) at the top, and the incorrect passive prediction in red at the bottom.

In the first row, we show samples where passive methods classify living entities as non-living entities, potentially leading to grave accidents in autonomous perception systems. The second row features samples where backgrounds have confused passive classifiers. In the third row, we present mispredictions that could be fatal for users. For example, in the second image of the third row, a passive vision system incorrectly classifying a garter snake as a harmless object could prove fatal for an individual relying on the system for navigation.

**Quantitative performance on ImageNet-A** In this section, we provide quantitative results supporting the advocacy of active vision for ImageNet-A. Figure 11 illustrates this effect. On the extreme left, a passive ResNet50 shows an accuracy of almost zero on this set. However, using the same ResNet50 as the ventral stream, FALcon improves performance to **5.32%** (Top) and **7.47%** (Any). For GFNet, this performance is **3.34%**. GFNet's slightly lower performance is due to its operation in a downsampled image regime, where images with a low object-to-background ratio experience significant feature shear.

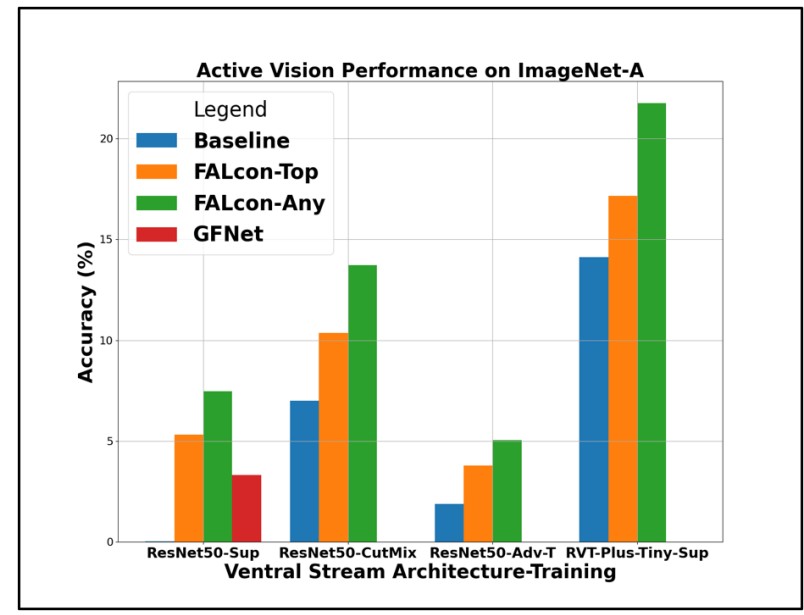

Previously, we observed that a strong ventral stream complements the dorsal stream for enhanced robustness against adversarially crafted samples. A similar phenomenon is again seen here. For CutMix-ResNet50 (Yun et al., 2019), Adv-T ResNet50 (Madry et al., 2018), and Robust Vision Transformer-Tiny (Mao et al., 2021), FALcon improves perfor-

Figure 11: Plot demonstrating the efficacy of Active Vision on ImageNet-A. The X-axis represents the architecture and training configuration of the ventral streams, while the Y-axis indicates accuracy across configurations. Active models with Supervised and CutMix pretrained ventral streams outperform those with Adv-T pre-trained ventral streams.

mance from **7.00%, 1.88%, 14.12%** to **10.37%, 3.79%, 17.16%** (Top) and **13.72%, 5.04%, 21.76%** (Any) respectively, improving the performance of all passive networks by **1.5x**. The efficacy of the "Any" predictions highlights the approach's strength for hard samples with complex scenes. Figure 16 in Appendix Section A.2 provides a qualitative intuition that aligns with the quantitative results.

**Key takeaway** Natural adversarial samples pose a significant challenge to passive classifiers. Active vision systems, with their ability to infer objects from various fixation points and make distinct multiple predictions, significantly improve the performance of underlying passive CNN and Transformer baselines. For ImageNet-A, we observe performance improvements even with Adv-T, unlike with adversarial samples, as adversarial training has less impact on naturally challenging samples compared to adversarially crafted ones. Notably, both ResNet50-Sup and ResNet50-CutMix show greater performance gains than ResNet50-Adv-T.

## 6 Foreground distorted images

**Human aligned active vision for foreground distortions** In this section, we qualitatively analyze the impact of foreground object distortions on the predictions of both active and passive vision systems. Unlike imperceptible adversarial noise that spreads non-uniformly across the image or naturally adversarial samples with varying degrees of adversity, these distortions visibly alter the structure of the foreground object.

Humans rely heavily on the structural configuration of objects to understand and identify them from various viewpoints. A visible deformation or distortion can significantly affect our confidence in recognizing an object. For instance, referring to each image in Figure 12, we may still be able to identify a golden retriever despite its parts being dispersed across the scene, though our confidence might waver. A clean image of a golden retriever might prompt a confident identification, whereas a distorted image might lead us to say, "I can see a golden retriever, but in parts dispersed across the scene." It is interesting to observe how DL-based systems predict these distorted adversarial images.

This section examines the response of these systems to such distortions and compares the performance of passive versus active processing. Our objective is to assess how well these systems align with human vision in handling visible structural distortions and to demonstrate how the human-aligned perception of an active vision system leads to more stable and interpretable predictions.

**Setup** The upper part of Figure 12 illustrates two methods for generating foreground-distorted images: (a) Image Shuffling and (b) Composite Images. For image shuffling, we divide an image into equal-sized patches and randomly shuffle them. If the foreground-to-background ratio is high, this process structurally distorts the object. For composite images, we use the Grab-cut algorithm (Rother et al., 2004) to extract the foreground, disassociate these parts, and paste them onto random backgrounds.

For evaluation, we use a passive ResNet50 and select FALcon as our candidate active vision model. Unlike GFNet, which predicts based on salient object parts, FALcon captures the entire object by gradually foveating on salient features and implicitly learning the structure, stopping when no further improvement is possible. This makes FALcon ideal for analyzing whether its predictions remain human-aligned despite foreground distortions. To comprehensively assess this, we conduct two-fold paired experiments: image shuffling and composite images. By concurrently applying these distortions, we aim to observe and compare the prediction trends of the two systems, thereby determining the robustness and human-alignment of their predictions.

**Qualitative analysis** For the undistorted sample of a golden retriever, both FALcon and ResNet50 make correct predictions with confidences of 75.39% and 41.56%, respectively. After **image shuffling**, FALcon makes multiple distinct predictions based on each localized part, each with lower confidence than the whole object. The face, being the most distinctive part, has the highest confidence among the parts. Low-confidence predictions ($< 50\%$) can be thresholded, as shown in Figure 12 (c). In contrast, the passive classifier's confidence abnormally increases, a phenomenon supported by literature (Tao et al., 2024; Chowdhury et al., 2024). This likely occurs because standard classifiers, trained on various image crops during data augmentation, rely on specific object parts for correct prediction rather than the overall structure. Thus, even when the image is shuffled and the structure is distorted, the discriminative parts still lead to strong predictions.

A similar instance is shown with a white wolf sample. For the second permutation of the image, the active method makes a categorical misprediction based on the localized facial part, identifying it as a "Russian wolfhound," which is still a "wolf-looking" dog breed. This misprediction is interpretable and less catastrophic, as evidenced by the provided sample of an actual Russian wolfhound. Image shuffling, the first of the two fold experiment indicates that passive methods rely mostly on parts for discrimination and lack structural

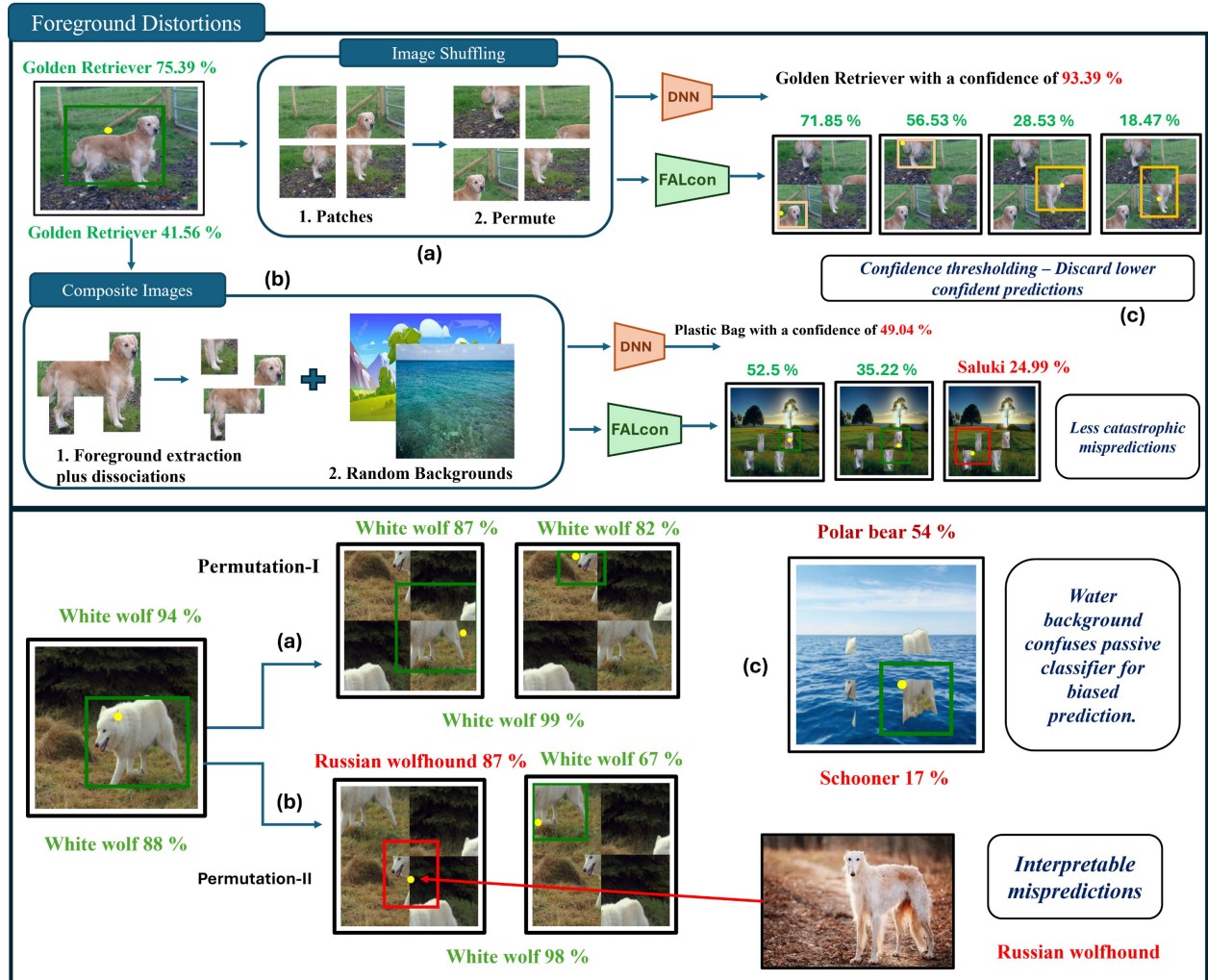

Figure 12: The figure illustrates two methods of creating foreground distortions and their effects on the predictions of active and passive vision systems. In (a) Image Shuffling, the image is divided into patches and then randomly permuted. In (b) Composite Images, the foreground is extracted from the original images, dissociated into parts, and then pasted onto random backgrounds. Across these two sets of distortions, active vision systems exhibit more stable, interpretable predictions and fewer catastrophic mispredictions compared to passive vision systems.

understanding. In contrast, active methods, while affected by distortions, make lower-confident correct predictions or interpretable, less catastrophic mispredictions.

In composite images, the same foreground is extracted, dissociated into parts, and then pasted onto random backgrounds, which are likely unfamiliar for passive classifiers to associate with the foreground. As seen in both samples, passive classifiers fail miserably, making predictions that are not remotely correlated with the actual foreground. On the other hand, active vision systems make lower-confidence correct predictions or interpretable, less catastrophic mispredictions based on the part localized by the dorsal stream. These predictions are also human-aligned; without seeing the original full image of a white wolf, humans might similarly infer that the parts belong to a wolf or a similar-looking dog breed like a Siberian husky or an Eskimo dog. For the corresponding composite image as well, the localized part resemble the body of a polar bear. Conversely, passive classifiers might make nonsensical predictions, such as identifying a polar bear's body part as a schooner, a type of sailing vessel, which is clearly a catastrophic error.

**Key takeaway** Thus, predictions indicate that across two different sets of foreground distortions, active systems showcase more resilient, stable, and human-aligned predictions. In contrast, passive systems, with their non-structured learning, and one shot way of inference, produce confident, erratic, and often catastrophic mispredictions.

## 7 Conclusions

In this work, we outline a deep learning-based active vision framework and advocate for its inherent robustness. Specifically, we adapt two existing approaches—FALcon Ibrayev et al. (2024b) and GFNet Wang et al. (2020)—into the active dorsal-ventral framework, demonstrating their robustness against three categories: adversarially crafted samples, naturally adversarial samples, and foreground-distorted images. In a black-box transfer attack setup, we attribute the enhanced robustness to two key factors: **(1)** glimpse-based processing at downsampled resolutions and **(2)** inference from multiple fixation points. Using GFNet, we show how downsampling mitigates the impact of adversarial noise. With FALcon, we demonstrate how multiple fixation points help avoid mispredictions due to the non-uniformity of adversarial noise. Employing various state-of-the-art adversarial transfer attacks, we observe consistent performance improvements of **2-21%** over passive methods, except when using non-adversarially trained ventral networks. We extend this understanding to natural adversarial samples, which model real-world challenges like object occlusions, unusual poses, and complex backgrounds that confuse passive classifiers. FALcon's flexible framework allows swapping different ventral streams, leading to a performance enhancement of **1.5x**, even with an adversarially trained ventral stream. We further investigate the robustness of image classifiers against foreground distortions, using FALcon as our active vision model due to its capability to capture entire objects in undisturbed images. Through a two-fold experiment, we visually demonstrate how the predictions of an active vision model are stable, resilient, and more human-aligned compared to the catastrophic mispredictions of passive classifiers. A potential future direction will be to explore optimized white-box attacks and defense mechanisms tailored to active vision systems. This exploration could lead to a deeper understanding of how to enhance these inherently robust systems, further strengthening their adversarial robustness.

**Acknowledgments**

This work was supported in part by, the Center for the Co-Design of Cognitive Systems (CoCoSys), a DARPA-sponsored JUMP 2.0 center, the Semiconductor Research Corporation (SRC), and the National Science Foundation.

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

# A Appendix

## A.1 Adversarially crafted samples

### A.1.1 DesNet121 as surrogate model

We provide additional results in this section with **DenseNet121** (Huang et al., 2017) as the surrogate model. $Acc_{s \to t}$ is provided as the metrics for comparison.

Table 4: Transferability results with DenseNet121 as Surrogate Model

| Surrogate | Target | PGD | VMI | PIFGSM |
|---|---|---|---|---|
| DenseNet121 | ResNet50 | 30.91 | 12.33 | 30.45 |
| | **FALcon-ResNet50-Top** | **43.19** | **20.0** | **35.70** |
| | **FALcon-ResNet50-Any** | **47.30** | **23.80** | **39.56** |

Table 4, presents results using adversarial samples transferred from DenseNet121 (Huang et al., 2017), an architecture distinct from the ResNet family. These results, evaluated across all 50,000 test samples of ImageNet with various iterative $L_{\text{inf}}$ norm attacks, demonstrate consistent improvements, further supporting our findings.

### A.1.2 Discussion with adversarially trained model

In this section, we briefly examine the role of an adversarially trained model (Madry et al., 2018) as a baseline in our analysis, alongside the robustness properties observed in an active vision system. We present results for both adversarially crafted samples (Section 4) and naturally adversarial samples (Section 5), summarizing key insights across various types of adversarial inputs.

- **Accuracy on non-perturbed clean samples** - The clean accuracy of an adversarially trained (Adv-T) model is lower than that of a standard classifier, a known trade-off resulting from adversarial training. This effect is evident in the clean accuracy results in Table 1.

- **Black box transfer attack setup** - In Table 1, we present Adv-T⋆ as an oracle method, trained to defend against worst-case perturbed samples within the L-∞ norm ball. The model shows minimal performance drop, even as the potency of adversarial attacks varies, with negligible decline compared to its nominal accuracy. Therefore, a black-box transfer attack setup is not the most effective approach for testing such a robust model.

- **Comparison with active vision systems** - In Table 1, we can see For attacks like PGD and MIM, GFNet outperforms Adv-T⋆, while FALcon-CutMix shows stronger performance on PGD and PI-FGSM attacks. However, active vision systems with non-robust ventral methods, such as CutMix ResNet50 and supervised ResNet50, as well as these passive ventral methods alone, experience a noticeable performance drop compared to clean accuracy, with results varying based on the attack's strength. This approach effectively highlights key performance trends for analysis.

- **Active vision system with Adv-T as ventral method** - In Figure 5 (a), for transferred attacks from LGV, we observe no noticeable improvements when adversarial samples are transferred from a surrogate supervised ResNet50, as highlighted in Section 4. However, for clean samples, there is a slight improvement.

- **Natural Adversarial samples** - In Section 5, we show that for naturally hard samples, a passive adversarially robust ResNet50 performs worse than a passive CutMix ResNet50. When integrated into an active setup, the framework improves performance on these samples; however, the gain is smaller than that achieved with CutMix or supervised ResNet50 in the same setup. This reflects the limitations adversarially trained models face with naturally occurring samples, a drawback not shared by inherently robust active vision systems.

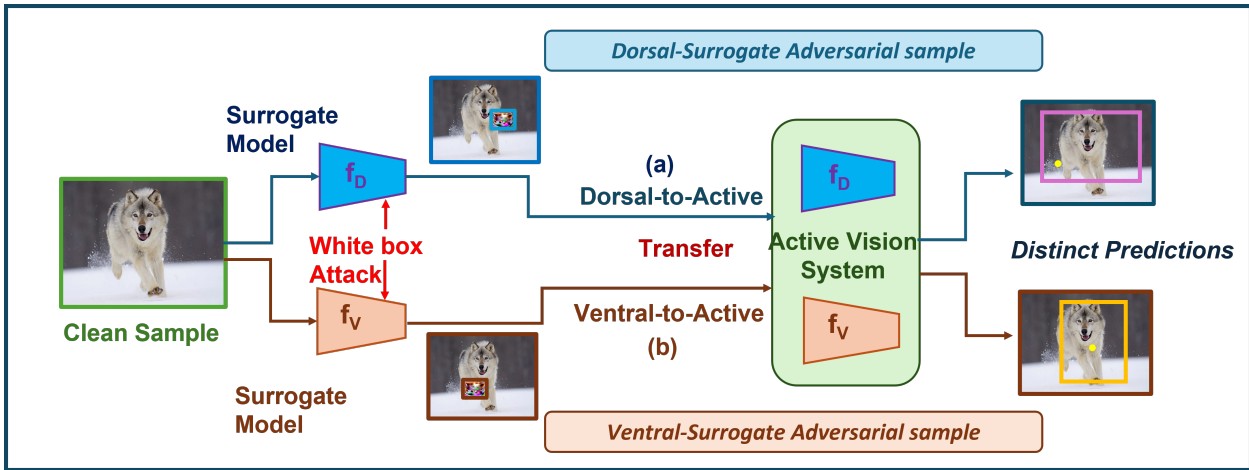

Figure 13: The figure illustrates the impact of generating adversarial samples using surrogate models with architectures matching either the dorsal or ventral networks. The adversarial sticker on each sample highlights the concentration of adversarial noise, depending on whether the dorsal or ventral network was used as the surrogate. "Dorsal-to-Active" (a) indicates an adversarial sample generated using the dorsal surrogate model and then transferred to attack the active vision system, with a similar setup applied for the ventral model (b). Only one transfer attack is generated each time. The distinct predictions reveal the variation in responses due to the different adversarial samples.

This discussion highlights that while an adversarially robust model serves as a strong baseline for adversarially crafted scenarios, it has drawbacks when applied to clean and naturally hard samples, as it is optimized for worst-case perturbations. In contrast, the bio-inspired active vision system demonstrates inherent robustness across diverse adversarially crafted and naturally hard samples, showing substantial gains by improving upon the underlying passive ventral stream. These two approaches are orthogonal: the adversarially robust model provides best-case performance on specific optimized inputs, while the active vision system delivers gains across a broader range of input types without being optimized for any particular kind.

### A.1.3    Transfer attacks illustration

This section details the setup outlined in Table 2, under Section 4. In a black-box transfer attack setup, the specific configuration of the underlying model remains unknown. However, to evaluate the efficacy of the active vision system, we generate adversarial samples by transferring attacks using surrogate dorsal ($f_D$) and ventral ($f_V$) streams. For FALcon, $f_D$ is represented by VGG16, and $f_V$ by ResNet50, as described in Section 4.2. This approach allows us to assess which stream serves as a more effective surrogate for adversarial sample generation. The lower section of Table 1 presents results when samples are crafted based on the ventral stream, ResNet50. Similarly, for Large Geometric Vicinity (LGV) (Gubri et al., 2022), transferred samples are generated using the surrogate ventral stream (ResNet50), with results illustrated in Figure 5 (a).

### A.1.4    Transferability from Large Geometric Vicinity (LGV)

The paper introduces a technique called Transferability from Large Geometric Vicinity (LGV) (Gubri et al., 2022) to enhance the transferability of adversarial attacks in black box transfer setup. This is illustrated in Figure 1 of (Gubri et al., 2022). The method starts with an initial pretrained surrogate model and gathers multiple weight sets for a few additional training epochs with a constant and high learning rate. This is done to enhance the geometric diversity of the surrogate models within a wide weight optimum. A wide weight optimum refers to a region in the weight space of a neural network where many configurations of weights result in similar, low loss values. In this region, the loss landscape is flatter or broader, meaning that small changes in the weights do not drastically increase the loss. Wide optima are often preferred because they represent solutions that are more likely to capture general patterns in the data rather than overfitting

to specific instances. This increases the likelihood of finding adversarial examples that are transferable to different models. Thus this approach leverages two geometric properties related to transferability:

- Wider Weight Optima: Models situated within a broader weight optimum serve as more effective surrogates.

- Effective Surrogate Ensemble Subspace: Identifying a subspace within this wider optimum facilitates the creation of an effective surrogate ensemble.

In our experimental setup in 4, we use ResNet50 as the surrogate model, matching the ventral stream of FALcon. Additionally, ResNet50 aligns with the feature encoders of GFNet, serving as feature extractors for both the ventral and dorsal streams. The results are presented in 5

### A.1.5 Token Gradient Regularization (TGR)

The TGR method, introduced in (Zhang et al., 2023) provides a gradient based transfer attack algorithm for Vision Transformers (ViT) (Dosovitskiy et al., 2021) and its variants such as Class-Attention in Image Transformers (CaiT) (Touvron et al., 2021) and Pooling based Vision Transformer (PiT) (Heo et al., 2021). This algorithm, represented as $TGR(\cdot)$ removes tokens with extreme values and reduces variance in back-propagated gradients. It utilizes token gradient information from both the Attention and Query-Key-Value components within an attention block, as well as from the MLP component within the MLP block, to generate adversarial samples $Grad_{adv}$. This is illustrated in Figure 1 of (Zhang et al., 2023). The TGR function combines gradient information as follows:

$$Grad_{adv} = TGR(Grad_{QKV}, Grad_{Att}, Grad_{MLP}, k, s)$$
$$x_{t+1}^{adv} = x_t^{adv} + \alpha \cdot \text{sgn}(Grad_{adv}) \tag{3}$$

Here, k denotes the top-$k$ or bottom-$k$ input gradients with highest and lowest values respectively which denote the extreme tokens. The scaling factor for gradients is $s$ and $\alpha$ is a hyper-parameter to control the step size. This method is effective against CNN models as well. And hence forms a strong transfer attack for both CNN and Transformer based backbones.

### A.1.6 Surrogate models for Token Gradient Regularization (TGR)

We follow the experimental setup of the original Token Gradient Regularization paper (Zhang et al., 2023) for selecting surrogate architectures. This paper demonstrated that transferable attacks, leveraging back-propagated gradients through attention blocks in specific surrogate vision transformers, are highly effective against other target vision transformer models. Additionally, it showed that transformer-based adversarial samples can successfully transfer to CNNs, making this approach effective for attacking CNN models as well. Following their setup, we chose a Vision Transformer (ViT) (Dosovitskiy et al., 2021) and its variants, including the Pooling-based Vision Transformer (PiT) (Heo et al., 2021) and Class-Attention in Image Transformers (CaiT) (Touvron et al., 2021), as surrogate models. We provide some intuition regarding the surrogate models.

- **ViT** - The Vision Transformer (ViT) architecture splits an image into fixed-size patches, treats each patch as a token, and applies a standard transformer model to these tokens, enabling direct application of transformer layers to image data without convolutional processing. In ViT-B/16, "B" stands for the Base model size, indicating a standard configuration with 12 transformer layers, and "16" refers to the patch size (16x16 pixels) into which the image is divided before processing.

- **PiT** - The Pooling-based Vision Transformer (PiT) modifies the Vision Transformer architecture by adding pooling layers between transformer blocks, which gradually reduce the spatial dimensions, similar to CNNs. This pooling improves efficiency and generalization. In PiT-B, the "B" stands for

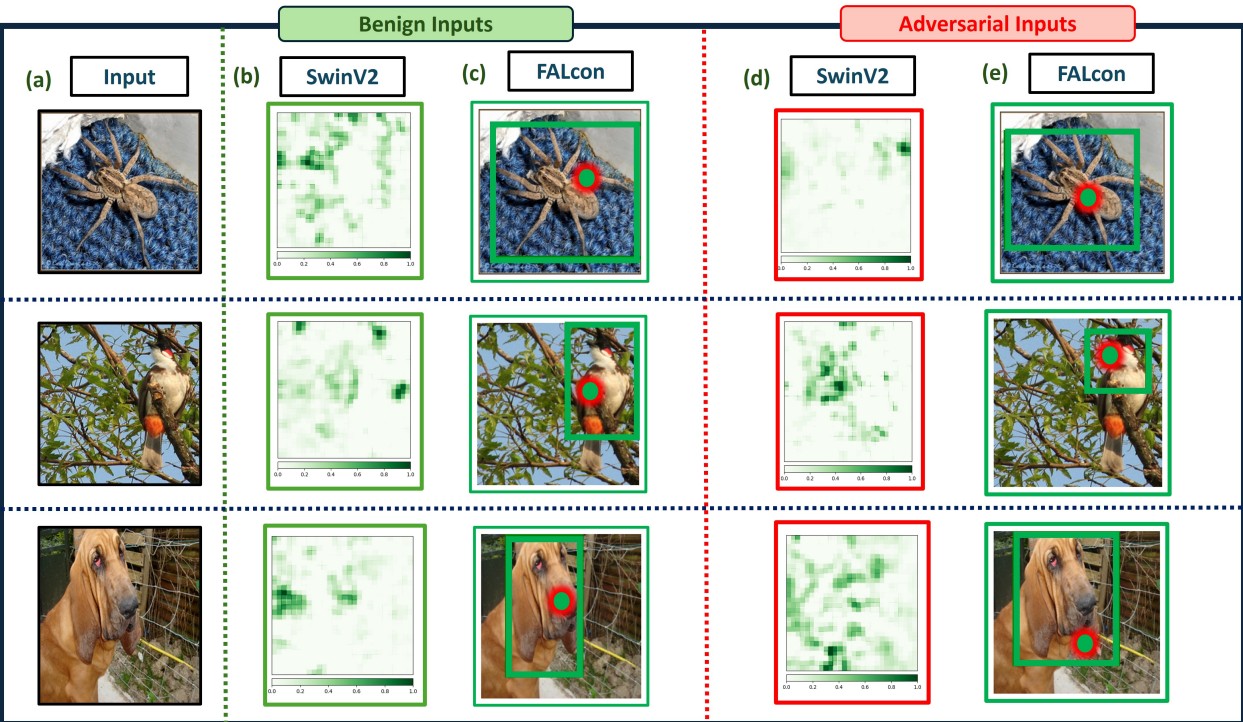

Figure 14: The figure illustrates the effects of adversarial noise from the TGR attack using the surrogate CaiT-S/24 transformer architecture. The target models include a passive, ventral-only Swin V2 transformer and an active vision model with FALcon's VGG16 dorsal and the same Swin V2 transformer ventral. Similar to Figure 8, this figure shows visualizations where the passive model is consistently misled, focusing on darker background regions rather than the intended foreground, as seen in occlusion maps under column (d). Although the active system is impacted by the adversarial noise as evident from the differing bounding box predictions compared to the benign inputs in column (c), it still guides the ventral stream to the correct classification in column (e).

Base model size, indicating a standard configuration with a specific number of layers and attention heads.

- **CaiT** - The Class-Attention in Image Transformers (CaiT) enhances the standard Vision Transformer by introducing class-attention layers at the end of the network, which focus on aggregating global information for classification, and LayerScale mechanisms within each transformer block, which help stabilize deeper models by scaling the outputs of layers for better training. In CaiT-S/24, "S" stands for Small model size, indicating a smaller configuration with fewer parameters, and "24" specifies the number of transformer layers in the network. This setup enables CaiT to go deeper while remaining stable and efficient.

While ViT-B uses 12 layers, CaiT-S/24 uses 24 layers, which makes CaiT-S/24 a deeper model despite being labeled as "Small" (S) due to its efficient configuration.

### A.1.7 Target models for Token Gradient Regularization (TGR)

In this subsection, we provide some intuition regarding the transformer based target models for the TGR setup Figure 5 (c). For transformer based target ventral models, we use Swin Transformer V2 (Liu et al., 2022), Robust Vision Transformer (Mao et al., 2021), and Focal Modulation Networks (Yang et al., 2022). And as mentioned in Section 4, we use the tiny versions of these transformers for evaluation.

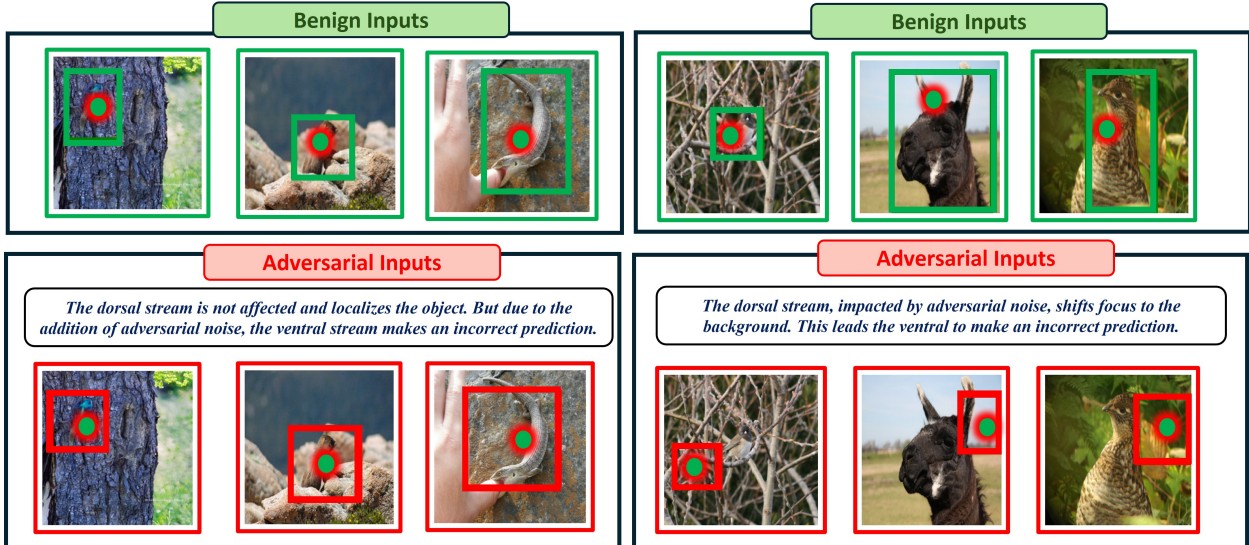

Figure 15: The figure highlights failure modes of a dual-stream active vision system. In both columns, the top row shows correct predictions on benign inputs, while the second row visualizes mispredictions. On the left, mispredictions occur when the ventral stream is impacted by adversarial noise, despite correct localization by the dorsal stream. On the right, adversarial noise affects the dorsal stream, shifting its focus to the background, leading the ventral stream to misclassify based on features from this misdirected region.

- **Swin V2** - The Swin Transformer V2 is an updated version of the Swin Transformer that processes images in shifted windows, enabling efficient hierarchical feature extraction. This design enhances image classification by effectively capturing both local and global features within the image.

- **RVT** - Robust Vision Transformer identified weaknesses in transformer models for adversarial robustness and introduced novel techniques, such as position-aware attention scaling and patch-wise augmentation, to enhance robustness across various shifts. These innovations make RVT a more resilient vision transformer, especially under adversarial and distributional shifts.

- **Focal Mod** - Focal Modulation Networks introduce a focal modulation mechanism that replaces the conventional self-attention approach, aligning more closely with human-like feature-based attention instead of spatial attention. This approach enhances the model's ability to learn aligned features, resulting in a stronger learned representation for various vision-based tasks.

For target ventral architectures in a transformer-based transfer attack, such as TGR, we selected an improved standard Vision Transformer in Swin V2, a robust vision transformer in RVT, and the Focal Modulation Network, which is inspired by neuroscience for enhanced feature representation. For each ventral model, we added FALcon's learned VGG16-based dorsal stream ($f_D$) to create active vision counterparts. FALcon-SwinV2-Tiny refers to a dual stream active vision system, with a SwinV2-Tiny ventral stream $f_V$ and FALcon's VGG-16 based $f_D$ dorsal stream. Figure 5 (c) illustrates these improvements, showing consistent trends similar to those observed with CNN-based transfer attacks and ventral backbones.

### A.1.8 TGR visualizations

We have included additional visualizations using TGR with CaiT-S/24 as the surrogate architecture. The target model comprises a passive Swin Transformer V2 Liu et al. (2022) and an active vision model combining FALcon's VGG16 dorsal stream with the same Swin V2 as the ventral stream. Figure 14 presents occlusion map visualizations similar to those in Figure 8, while Figure 15 illustrates failure modes critical to understanding active system behavior. Figure 14 demonstrates a generalization of the visualizations from CNN-based iterative attacks on CNN targets to a similar transformer-based scenario, showing consistent effects in this setup.

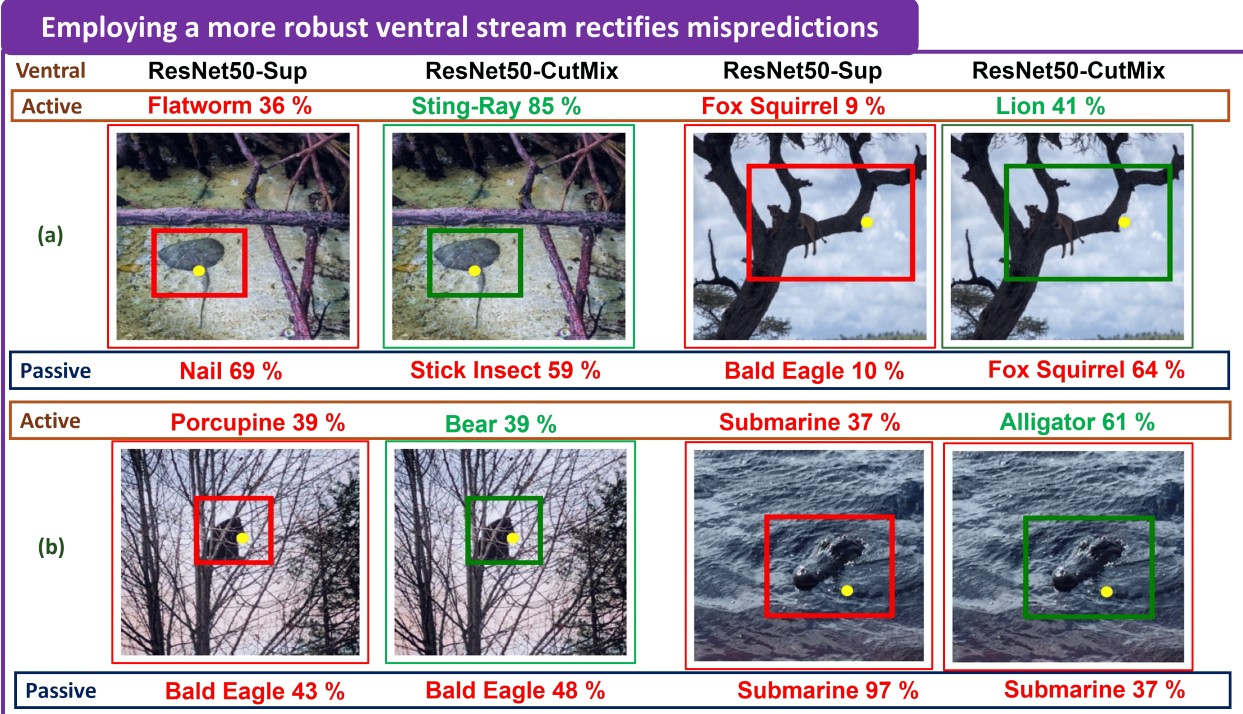

Figure 16: The dorsal stream captures the object in all cases, and a better ventral stream improves upon the predictions. The ResNet50-CutMix ventral stream rectifies mispredictions made by a ResNet50-Sup ventral stream. The quantification of this effect is shown in the bar plot on in Section 5.

## A.2    Natural adversarial Images

The figure 16 shows qualitative results, with ventral stream training configurations and passive incorrect predictions displayed at the bottom of each image. CutMix, being a more robust ventral stream than a standard ResNet50, improves predictions, with the stable dorsal stream correctly localizing objects in all cases.

