# On Inherent Adversarial Robustness of Active Vision Systems Supplementary

**Amitangshu Mukherjee**                                        *mukher44@purdue.edu*
*Elmore Family School of Electrical and Computer Engineering*
*Purdue University*

**Timur Ibrayev**                                               *tibrayev@purdue.edu*
*Elmore Family School of Electrical and Computer Engineering*
*Purdue University*

**Kaushik Roy**                                                 *kaushik@purdue.edu*
*Elmore Family School of Electrical and Computer Engineering*
*Purdue University*

**Reviewed on OpenReview:** *https://openreview.net/forum?id=iVV7IzI55V*

## 1  Active Vision Systems

In this section, we offer a high-level overview and insights into the learning of the two candidate active vision methods — Glance and Focus Networks (GFNet) Wang et al. (2020) and FALcon Ibrayev et al. (2024).

### 1.1  FALcon

**Active Vision structure :**  Both the dorsal $f_D$ and ventral $f_V$ streams are represneted by deep convolutional neural networks. During training, only $f_D$ is trained to emulate the saccadic and foveated functions. For $f_V$, any pre-trained network can be selected.

**Learning :**  In the upper part of Figure 1, we present a high-level overview of the learning stage of FALcon. The learning algorithm directs a dorsal network $f_D$ to produce a sequence of foveated glimpses using two learned techniques: foveation and saccades. An initial fixation point marks the starting position of the expansion process. A foveated glimpse is defined as a cropped region from the image (initiated at fixation points) with dimensions dictated by the foveation expansion process, as described later. Each foveated region is downsampled to lower dimensions of $H' \times W'$. For localization, given an input image $X$, an initial fixation point and a pseudo-bounding box (illustrated as an orange dashed box), the dorsal $f_D$ learns five distinct actions, as depicted in the diagram, to produce a sequence of foveated glimpses on which the network is trained. This is a sequential iterative process carried out for $T$ steps. The glimpse at the last step of the iterative process is defined as the final foveated glimpse. The dimensions of the final foveated glimpse are designated as the dimensions of the predicted bounding box.

The dorsal begins its process with an initial fixation point (depicted as a red point) within the orange pseudo bounding box. Starting with a predefined resolution for the initial glimpse, the dorsal then iterates through four potential expansions: rightward $(dx^+)$, leftward $(dx^-)$, downward $(dy^+)$, and upward $(dy^-)$, with the top-left corner of the initial glimpse as the origin $(0,0)$. These expansions aim to emulate the foveation process, wherein the model actively adjusts its focus to capture salient object features. The model's decision to expand in each direction is based on its confidence in capturing an object, with expansions guided by a fixed glimpse step size. Importantly, the expansions are constrained by the pseudo bounding box and the

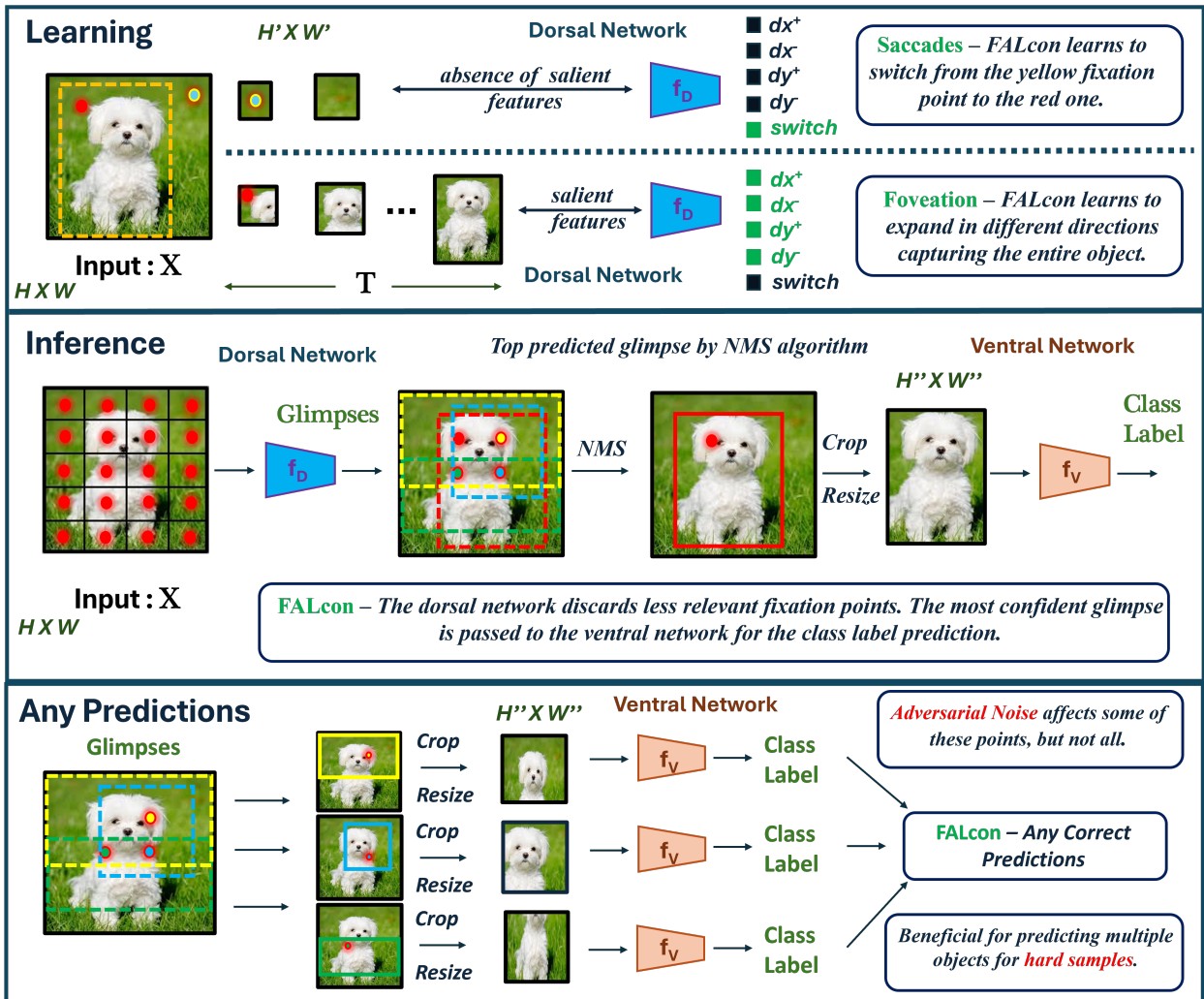

Figure 1: The figure provides a high-level overview of FALcon. During **Learning**, the dorsal network ($f_D$) is trained to predict five distinct actions (four for expansion and one for switching), enabling it to learn the importance of each fixation point illustrated by colored dots. Learning occurs in a downsampled resolution of ($H' \times W'$). During **Inference**, $f_D$ starts from each pre-defined multiple fixation point (20 red dots). If salient object features are present, $f_D$ performs the learned expansions to capture the object (4 colored dashed boxes, colored dots). The most confident final foveated glimpse (red solid box) is cropped ($H'' \times W''$) and presented to the ventral ($f_V$) for **Top** prediction. The system can produce distinct predictions referred to as **Any**, which are beneficial for handling adversarial samples.

relative position of the initial fixation point to prevent glimpses from extending beyond this boundary. At each iteration $t$, the target for expansion is determined based on the dimensions of the current glimpse and the reference pseudo-bounding box. For instance, if the target is $(1, 0, 1, 0)$, the model prioritizes expanding to the right and downward given the bounding box's location and the current glimpse. This iterative process continues for a fixed length of the foveation sequence, denoted as $T$, until the dorsal successfully captures the entire object.

During training, FALcon undergoes a learning process where it evaluates the relevance of a fixation point and decides whether it needs to switch to another fixation point. If the initial fixation point falls outside the pseudo bounding box (orange point), indicating a potential need for a switch to focus on relevant features, the network is guided to make the switch (switch target is *true*). Conversely, if the network determines that a switch is necessary based on the absence of salient features in the current glimpse sequence, a new fixation point within the pseudo box is presented, and no switch is required (switch target is *False*). This

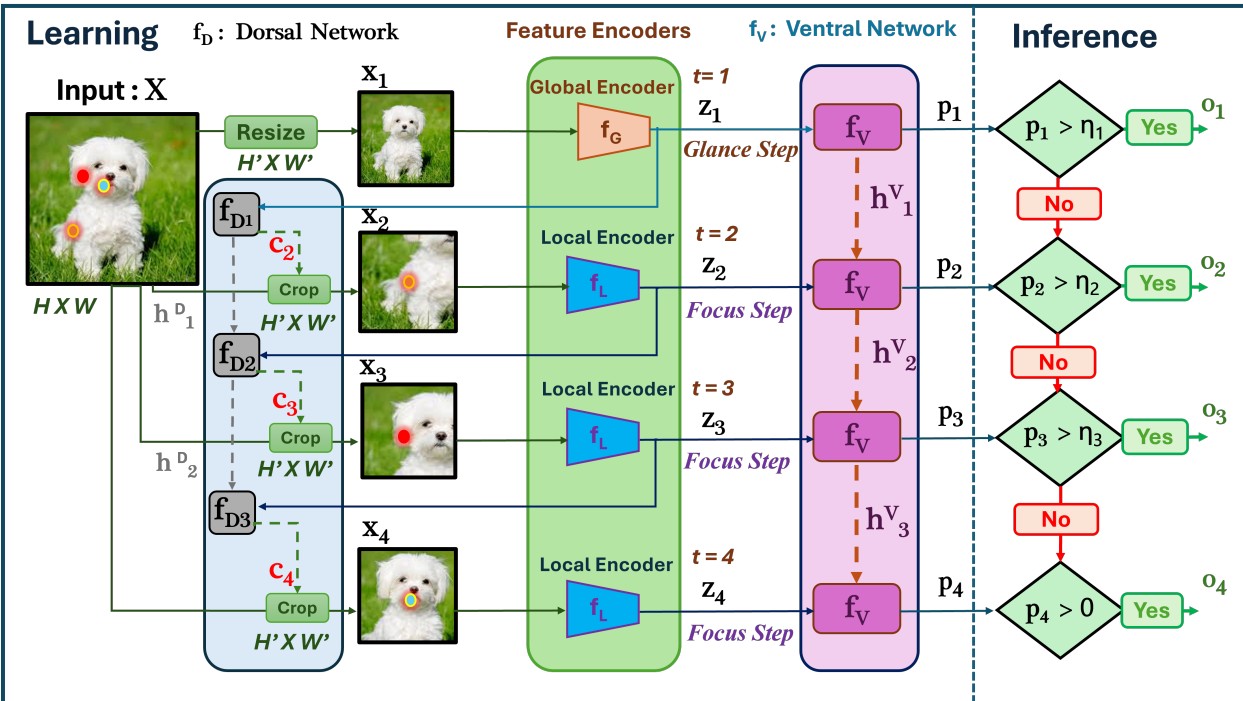

Figure 2: (**Learning & Inference**) The figure provides an overview of GFNet's operation. It begins by downsampling the input image to a lower resolution for rapid prediction ($p_1$), termed Glance at $t = 1$. If the network lacks confidence ($p_1 < \eta_1$), it enters subsequent Focus steps until certainty is attained or till ($t = 4$). Each focus step analyzes a patch ($H' \times W'$) cropped from the original input ($H \times W$) centered around ($c_t$) illustrated by colored dots. These co-ordinates are determined by the dorsal network $f_D$. The process is depicted for a sequence length of 4.

learning mechanism simulates the way human vision actively adjusts focus, akin to saccadic movements when essential features are not detected in the current field of view.

By learning the relevance of different fixation points, FALcon learns to capture objects using various trajectories of foveated regions. This ensures the diversity of learned representations of salient object features in an image. The dorsal is a deep convolutional neural network, and the five actions are trained based on binary cross-entropy loss between predictions and targets per iteration $t$ in the input sequence of length $T$.

## 1.2   Glance and Focus Networks

**Active Vision structure :** GFNet potrays a slightly more complex framework as DL based active vision system. Both the dorsal $f_D$ and ventral $f_V$ streams are represented as deep recurrent neural networks to aggregate information from previous steps. For processing image features, both streams employ convolutional deep neural network based feature encoder backbones $f_{G/L}$. In the training phase, all networks are trained.

**Learning :**   The learning occurs in two distinct steps – a glance step and subsequent multiple focus steps. In the diagram, we illustrate the learning till $T = 4$ which is the maximum length of the sequence. Each subsequent step is denoted by $t$. The task at hand is to correctly classify an input image $X$ within a given number of time steps.

In the first step, the image with full resolution ($H \times W$) is downsampled to a much lower resolution ($H' \times W'$) where $H' < H$ and $W' < W$, and then processed to make a quick prediction based on the globally processed information through a global encoder $f_G$ and ventral stream $f_V$ pathway. This is known as the *Glance* Step, akin to how humans tend to make quick predictions by glancing through images. Along with the prediction $p_t$, a dorsal network $f_D$ predicts a region proposal based on the most class-distinctive region for the subsequent step ($t + 1$) to process. The output of this network consists of the location of each image

patch, which comprises normalized image coordinates corresponding to the center coordinates of each patch $(c_{t+1})$. These image co-ordinates are the fixation points for the subsequent glimpses.

At $t = 2$, as illustrated in Figure 2 there is an orange fixation point $(c_2)$ around the leg region of the dog which is the predicted co-ordinate at that time step. Using these coordinates, the corresponding patch is cropped from the full image at size $H' \times W'$ and fed to the local encoder $f_L$ and $f_V$ pathway. Since the cropped image is a small patch entailing sharp details, this step is known as *Focus* step. This step is repeated until the maximum length of the input sequence, which is $T = 4$ in this case. During each iteration, the local encoder is progressively trained on the smaller patches based on the fixation points determined by the dorsal network. The sequential glimpses are illustrated in the diagram, centered around distinct fixation points per step as given by $c_2, c_3$, and $c_4$ for time steps $2, 3$, and $4$ respectively. Thus, at each focus step, the network observes different class-discriminative regions of the object with sharper details to successfully classify the image.

The global and local encoders $f_G$ and $f_L$ respectively are deep convolutional neural networks and the ventral $f_V$ is a deep recurrent neural network. The ventral $f_V$ aggregates the information from all previous inputs $(h^c_{t-1})$ and the subsequent feature maps $z_t$ from the encoder to produce a prediction at each step. $f_G$, $f_L$ and $f_V$ are trained simultaneously in a supervised manner to produce correct predictions $p_t$ at high confidences for the entire length of input sequence from $t = 1$ to $t = 4$ as depicted in the diagram.

The patch dorsal network is also a recurrent neural network trained via a policy gradient algorithm to select the patches that maximize the increments of the softmax prediction on the ground truth labels between adjacent two steps. The inputs to this dorsal network are the previous hidden representations and subsequent feature maps, denoted as $h^D_{t-1}$ and $z_t$ respectively.

Thus, the network learns the representations of each object by actively predicting class-discriminative image patches centered around the fixation points and learning the same, which consist of important and salient details.