# OpenReview forum: "On Inherent Adversarial Robustness of Active Vision Systems"
_TMLR — Accepted by TMLR_

### Review · Reviewer_6bhc · 2024-10-13

**Summary Of Contributions:**

Summary: This paper works to design an inherently adversarially robust artificial vision systems using inspiration from biological vision. Specifically, they implement an artificial version of dual stream dorsal (where) and ventral (what) vision system in which images are processed as a series of random saccades. This can naturally handle threats involving distractor objects. Their implementation based on FALcon and GFNet incorporates foveation and saccades.

**Audience:**

Yes

**Broader Impact Concerns:**

Concerns: None

Overall thoughts: I think that this paper is clearly of a high quality. With some minor changes, I believe that it would be ready for publication. I think that it will be valuable to the vision and robustness fields. Unfortunately, I have not followed vision robustness research closely in the past two years, but based on my knowledge, this paper seems like it will provide distinct value.

**Claims And Evidence:**

Yes

**Requested Changes:**

Minor: consider citing [https://arxiv.org/abs/1511.06292](https://arxiv.org/abs/1511.06292)

Minor: Consider adding a figure to illustrate the attacks in table 2.

Major: Figure 3 is almost unreadable because of text size. Also figure 11a.

Medium: I think that this paper does enough. But I also idealistically wish that somewhat more had been done to find flaws with their methods -- perhaps attacks optimized under transformations (e.g. foveation) and that targeted salient features?

**Strengths And Weaknesses:**

S1: I think is is excellent how this paper tests against adversarial examples in the form of pixel perturbations, natural adversarial images, and foreground distortion. I also think that it is great that CNNs and transformers were used.

S2: I find the paper to be very thorough and to have high quality experiments. I am struggling to think of ideas for things to include that were not already done. Nonetheless, see below.

S3: It's well-written with high-quality figures

Weaknesses: I think that the right way to understand this paper is as a basic science one. I'm reviewing it as such. But it would have also been nice to compare and/or compete with SOTA adversarial training and randomized smoothing baselines.

---

> ### Author Response · Authors · 2024-11-09
> **Rebuttal for Reviewer 6bhc**
>
> We thank Reviewer 6bhc for their supportive comments and for highlighting the key strengths of our paper, particularly the thoroughness of the experiments.
>
> ### Requested changes
>
> > We have added the suggested citation -- thanks for the recommendation.
> > We have added Figure 13 in the Appendix to illustrate the setup related to Table 2, and have referred it accordingly.
> > We have increased the resolution of Figure 3. Additionally, we have separated Figure 11, retaining only the plot (previously Figure 11(a)) that presents quantitative results on ImageNet-A. The corresponding qualitative illustration, which aligns with this plot, have been moved to the Appendix (A.2) and is referred in Section 5.
>
> ### Addressing Medium Changes - Attacks Optimized under Foveation
>
> >We appreciate your insightful suggestion regarding attacks optimized under glimpse-based foveation, as it would indeed be a valuable test for evaluating the resiliency of such systems. However, this type of attack falls under a white-box setup, requiring access to the forward model of active vision systems to precisely perturb the regions targeted by the dorsal stream. Designing a white-box attack that accurately affects each forward model, such as FALcon or GFNet, represents an area for future research in active vision. In our black-box transfer setup, however, we generated attacks based on the ventral and dorsal streams separately for FALcon, as shown in Table 2. Figure 13 provides a visual intuition behind this. Thank you for the feedback.
>
> ### Comparison with Adversarially trained baselines
>
> We have updated Table 1 to include an adversarially trained ResNet50 as a baseline. In Sections 4.2 and 5, we emphasize this baseline’s role for LGV and natural adversarial samples, with highlighted portions already present in our initial submission. Additionally, we have added a discussion in Appendix A.1.2 to further elaborate on the role of an adversarially trained model as a baseline in our analysis.
>
> >**Adversarial crafted samples** - As Reviewer 6bhc rightly noted, our primary goal is to demonstrate how the dual-stream architecture inherently boosts robustness of the underlying ventral stream. For ventral models already trained on worst case $L_{\inf}$ perturbations, our dual-stream approach does not provide a strict improvement compared to other ventral methods, a point we emphasize and highlight in under **Transfer Attacks with Large Geometric Vicinity (LGV)** on Page 10 in purple.
>
> >**Clean samples** - Additionally, adversarial trained models often show decreased performance (clean accuracy) on natural benign images—a trade-off our approach avoids.
>
> >**Natural Adversarial samples** - Section 5 shows that, for naturally adversarial images, our active vision system improves the performance of the underlying adversarial trained model as well. Notably, active vision with non-adversarial trained ventral methods, like CutMix ResNet50 and supervised ResNet50, outperform Adv-T ResNet50 models in this setting. This highlights that, while adversarial training is effective for adversarial perturbed images, it lacks versatility for other types of adversarial challenges.

---

> > ### Comment · Reviewer_6bhc · 2024-11-10
> > **I think the paper should be accepted**
> >
> > Thanks to authors for a very clear and comprehensive response and update to the paper. I have no outstanding concerns with the paper, and I think it will be of value and interest to the TMLR, ML, and ophthalmology communities.

---

### Review · Reviewer_zPwx · 2024-10-20

**Summary Of Contributions:**

This paper discusses the adversarial robustness of active vision systems. These active vision systems are inspired by neuroscience and perform identify features by “actively switching between multiple fixation points”. This happens by using two pathways: the dorsal pathway, that focusses on where the object is and the ventral pathway which is used to classify the object. Two existing active vision methods are used in the paper: FALcon and GFNet. The authors analyse how robust these active vision models are to adversarial attacks (in detail, transfer attacks that calculate an adversarial perturbation on a surrogate model). It is shown that these active vision models are somewhat more robust to adversarial examples than deep learning models that process the image in a single forward pass (e.g., the ResNet). Furthermore, the robustness to natural adversarial images and foreground distorted images is discussed.

**Audience:**

Yes

**Broader Impact Concerns:**

No concerns.

**Claims And Evidence:**

No

**Requested Changes:**

- Address weaknesses (W1.1)-(W1.5).
- Address weakness (W2)
- Further polish the paper (W3). Here are some suggestions on how to do this:
    - The figures are very helpful in explaining the work but have very low resolution. Also, captions are sometimes to small. There are typos („mis prediction“) and some figures can be further improved (e.g., the frames of the heatmaps in Figure 8 look uneven)
    - The definition of the accuracy (Equation 2) is wrong. Here Iverson brackets or an indicator function should be used around the expression $T_i(x_{advj})=y_j$. Also, $x_{advj}$ is a bad choice for a variable name. Furthermore, the $Acc_{s\rightarrow t}$ should be included in the equation. What is meant with $s\rightarrow t$?
    - Fix the following typographic issues:
        - Capitalization of section titles is somewhat inconsistent (e.g., “Natural Adversarial images”). Also, words like “Active Vision” or “Active Localization” are sometimes capitalized for no reason.
        - Citations are wrongly formatted almost everywhere: I suggest authors use the Latex commands `\citep{}` and `\citet{}` to properly include the citations in the sentence.
        - Paragraph titles are sometimes followed by a “:”.
        - Optional: It would be better if references mention the long conference name instead of the abbreviation (e.g., ICML).
    - Table 2 is too wide.

**Strengths And Weaknesses:**

Strengths:

- The findings are interesting but not too surprising. However, I see a that the community could profit from these experiments.
- The paper is well-written and I like the short key takeaway sections very much.
- The authors extensively describe necessary background in active vision systems (i.e., the FALcon and GFNet architectures and their biological motivation) which makes the paper self-contained.

Weaknesses:

- (W1) In my opinion the performed experiments could be more detailed to further back the claims made in the paper:
    - (W1.1) The main result table (Table 1) are interesting but I would suggest that the authors evaluate also evaluate a whitebox attack scenario where adversarial perturbations are directly calculated for the model. It would be interesting if the robustness claims made by the authors also hold in this setting.
    - (W1.2) The chosen attack methods discussed in Section 4.2 is debatable. In my opinion, there is too much focus on FGSM/PGD attacks and their variants. I am missing evaluation of other well-established attacks like the Carlini-Wagner attack [1] or universal adversarial perturbations [2].
    - (W1.3) Table 1 shows ResNet50 perturbations evaluated on a ResNet50 architecture (target and surrogate model are the same). Would it not be more interesting to evaluate the ResNet50 perturbations on a ResNet34 architecture instead?
    - (W1.4) Results in Section 5 and 6 are only backed by examples and is missing an extensive quantitative evaluation.
    - (W1.5) The title of the paper is in my opinion an exaggeration. The authors show that active vision systems are slightly more robust than passive models but not that they are completely robust to adversarial attacks. I would suggest that the authors consider a title that describes what is done in the paper and is less flashy.
- (W2) Add some sentences that explain the intuition for the used transfer attacks (LGV and TGR).
- (W3) Overall, the paper needs further polishing before it can be published. The type-setting can be improved at many places (see requested changes for some examples).

References:

[1] Carlini, Nicholas, and David Wagner. "Towards evaluating the robustness of neural networks." *2017 IEEE symposium on security and privacy (SP)*. IEEE, 2017.

[2] Moosavi-Dezfooli, Seyed-Mohsen, et al. "Universal adversarial perturbations." *Proceedings of the IEEE conference on computer vision and pattern recognition*. 2017.

---

> ### Author Response · Authors · 2024-11-09
> **Rebuttal for Reviewer zPwx**
>
> We thank Reviewer zPwx for their thoughtful comments and positive feedback. We are glad to hear that the findings and key takeaway sections were valuable and that the background provided on active vision systems helped make the paper self-contained. We appreciate the reviewer’s recognition of the potential benefit these experiments offer to the research community.
>
>
> ### Requested Changes  - W2 and W3
>
> > **W2** – Thank you for the suggestion. We have added sections A.1.4 and A.1.5 to explain the intuition behind the transfer attacks, including Large Geometric Vicinity for CNNs and Token Gradient Regularization for Transformers.
>
> > **W3** – We appreciate the helpful feedback on polishing the manuscript.
> > > We have refined the definition in Equation 2 by incorporating the indicator function, and clarified that s → t denotes surrogate to target. Thank you for the suggestion.
>
> > >  We have enhanced the figure resolutions, corrected the citation format, and addressed the other suggested changes in the updated version of the manuscript.

---

> > ### Author Response · Authors · 2024-11-09
> > **Addressing Weakness (W 1.2 and W 1.3)**
> >
> > >**W1.2** – Thank you for the suggestion to include Carlini-Wagner (C&W) as an additional attack method. While the C&W attack is a powerful white-box method under the $L_{2}$ norm, it is generally less effective as a transfer attack compared to $L_{\inf}$-norm based methods like PGD [1]. This is due to the sharper, localized perturbations of $L_{\inf}$ attacks, which tend to transfer more effectively across models than the smoother $L_{2}$ based perturbations used in C&W. $L_{2}$ based approach creates more diffused changes across the image, often making it less transferable. For this reason, we focused on PGD and its variants in the current black-box transfer attack setup.
> >
> > >We demonstrate this in the table below, where we generate adversarial samples on 1,000 randomly sampled images from the ImageNet test set using C&W, PGD-$L_{2}$, and PGD-$L_{\inf}$ attacks. The higher accuracy for C&W and PGD-$L_{2}$attacks indicates their lower transferability in a black-box transfer attack setup. This trend is evident for both the passive ventral-only baseline (ResNet34/50) and FALcon. Notably, FALcon enhances the performance of the underlying model across all transfer settings, aligning with our previous findings.
> >
> > |   Surrogate             |Target             |CWL2      |PGD-L2  |PGD-L-$\inf$             |
> > |----------------|-------------------------|----------|--------|-----------|
> > |ResNet34				 |ResNet50                 | 67.60    | 61.50  | 29.60     |
> > |                |FALcon-ResNet50-Top      | 69.10    | 66.20  | 47.40
> > |                |FALcon-ResNet50-Any      |71.30     | 68.0   | 51.60
> > |ResNet50				 |ResNet34                 |   66.50  | 61.30  |    34.10  |
> > |                |FALcon-ResNet34-Top      |   67.90  | 64.60  | 45.60
> > |                |FALcon-ResNet34-Any      |   69.80  |  66.30  | 50.50
> >
> >
> > > **W1.3** – We appreciate the reviewer’s suggestion to evaluate ResNet50 perturbations on a ResNet34 architecture for additional insight. The bottom part of Table 1 was specifically designed to emphasize the effect of transferred samples from an architecture identical to the ventral model in our active vision system (FALcon). This setup demonstrates that, even when the target and surrogate share the same architecture, adding a dorsal stream shields the ventral stream, making the overall system inherently more robust.
> >
> > >To address this suggestion, we also provide results in the table evaluating ResNet50-generated perturbations on a ResNet34 architecture and FALcon with a ResNet34 ventral stream, where we observe similar trends in robustness. Additionally, we provide results using adversarial samples transferred from DenseNet121 in Appendix A.1.1, an architecture distinct from the ResNet family. These results, evaluated across all 50,000 test samples of ImageNet with various iterative $L_{\inf}$ ​-norm attacks, show consistent improvements, further supporting our findings. This is a good suggestion, as further cross-architecture analysis could help identify key robustness aspects of the active vision system.
> >
> > |   Surrogate            |Target             |PGD      |VMI  |PIFGSM             |
> > |----------------|-------------------------|----------|--------|-----------|
> > |DenseNet121			 |ResNet50                 | 30.91    | 12.33  | 30.45     |
> > |                |FALcon-ResNet50-Top      | 43.19    | 20.0  | 35.70
> > |                |FALcon-ResNet50-Any      |47.30     | 23.80   | 39.56
> >
> > 1. **Tramèr et al. (2017)** - _"The Space of Transferable Adversarial Examples."_

---

> > > ### Author Response · Authors · 2024-11-09
> > > **Addressing Weakness (W 1.4)**
> > >
> > > > **W1.4** – For the natural adversarial samples in Section 5, we provide quantitative results on the ImageNet-A benchmark for both CNN- and Transformer-based ventral methods, as shown in Figure 11 and explained in  subsection, titled “Quantitative Performance on ImageNet-A".  Figure 14 in Appendix Section A.2, further illustrates the impact of a stronger ventral stream, complementing the quantitative findings in Figure 11.
> > >
> > > > The qualitative results highlighted in Figure 10 demonstrate why ImageNet-A is challenging, as it includes factors such as complex backgrounds and occlusions that can hinder performance for passive ventral-only methods. The explanation is provided in subsection titled as "Catastrophic mispredictions by Passive classifiers". Active vision, however, performs effectively in these challenging scenarios, as shown in our qualitative analysis.
> > >
> > > >  The primary focus of Section 6 was to visually demonstrate how the predictions of active vision systems align with human perception. For this purpose, we conducted a two-step experiment on a few samples from the ImageNet dataset to assess prediction stability under visible foreground distortions. Our goal as demonstrated in Figure 12 was to illustrate that the active system’s predictions remain consistent under these distortions, reflecting a stability similar to human perception. In contrast, passive methods that process the entire image tend to show erratic predictions across different foreground distortions. In this section, we made only qualitative claims, as this experiment is centered on understanding semantics and syntax within images, a potentially new area [2] without a clear benchmark for quantitative evaluation.
> > >
> > > 2. **Tao et al. (2024)** - "Towards Image and Syntax Sequence Learning"

---

> > > > ### Author Response · Authors · 2024-11-09
> > > > **Addressing Weakness (W 1.1 and W 1.5)**
> > > >
> > > > > **W1.1** – While we appreciate the reviewer’s suggestion, we have not made robustness claims under a white-box attack scenario. Our focus is on demonstrating the inherent resilience of the active vision system within a black-box transfer setup. Access to gradients would be essential for directly perturbing regions targeted by the dorsal stream. Conducting a true white-box attack on the active vision system would require the full forward model, including the iterative dorsal stream over multiple fixation points and the ventral classifier. Developing such a white-box attack for models like FALcon or GFNet would indeed be an interesting direction for future research in active vision, though it lies beyond the scope of our current work. However, we have already demonstrated the effects of **transferring** adversarial samples based on either the ventral or the dorsal stream.
> > > >
> > > > >>  The bottom part of Table 1 shows a scenario where adversarial samples are generated using a surrogate ResNet50, the same architecture as the ventral stream, and transferred to attack our target models. This setup demonstrates that, even if an attacker knows the ventral stream architecture (not assumed in a black-box setting), adding a dorsal stream enhances resilience to transferred attacks. Architectural details are provided in Section 4.1, **Implementation Details**.
> > > >
> > > > >> In Table 2, we further extend this approach by using AutoPGD to generate adversarial samples with surrogate models for both the dorsal (VGG16) and ventral (ResNet50) architectures in FALcon. These results show that attacks generated based on the ventral stream architecture are notably more effective in this context. For added clarity, we have included a pictorial illustration of this setup in Section A.1.3.
> > > >
> > > > > **W1.5** – We appreciate the opportunity to clarify our intent. We do not claim in the paper to propose a comprehensive adversarial defense system. Instead, we highlight that the active vision system exhibits robustness properties similar to those observed in human vision, where robustness is inherent. This robustness is a natural result of the dual-stream structure, with the dorsal stream guiding the ventral stream, which adds resilience to different kinds of adversarial inputs. Additionally, we do not claim complete adversarial robustness, which would typically involve adversarial training and white-box testing. Our goal was to showcase the inherent robustness characteristics of the active system against three types of adversarial samples in a black-box transfer attack setup. We hope this explanation clarifies our position and the scope of the claims made in the paper.

---

> > > > > ### Comment · Reviewer_zPwx · 2024-11-11
> > > > >
> > > > > **W1.4:** Thank you for your clarification!
> > > > >
> > > > > **W1.1 and W1.5:** Thank you for clarifying your intent. In my opinion, the results are worth publishing with TMLR, but at this point the issue still persists. The paper is titled "On Inherent Adversarial Robustness of Active Vision Systems" which is in my opinion way too general. A potential reader would expect an in-depth analysis of the adversarial robustness of such systems, which would also include white-box attacks because this attack scenario is the most fundamental one discussed in literature. I would recommend to clarify the focus of this work in the title of the manuscript.

---

> > > > > > ### Author Response · Authors · 2024-11-12
> > > > > > **Response to title change recommendation**
> > > > > >
> > > > > > To address your suggestion, we are open to modifying the title to "On Inherent Robustness of Active Vision Systems".

---

> > > > > > > ### Comment · Action_Editor_Vury · 2024-11-17
> > > > > > > **W1.1 and W1.5**
> > > > > > >
> > > > > > > Dear zPwx,
> > > > > > >
> > > > > > > thank you very much for the active engagement with the authors. It looks like there is willingness on both sides to resolve the remaining issue, e.g., the authors have offered to change the title of the manuscript.
> > > > > > >
> > > > > > > To me it sounds like a clear statement (in the introduction / limitations section) of the scope of the paper, maybe in addition with a changed title could solve the remaining issue. If you agree, zPwx, please state this briefly and be as specific as possible w.r.t. which concrete changes would sufficiently address your open criticism.
> > > > > > >
> > > > > > > To the authors: please consider if changes/additions could help clarify the scope of the work even further.
> > > > > > >
> > > > > > > Thank you all for the active engagement!

---

> > > > > > > > ### Comment · Reviewer_zPwx · 2024-11-17
> > > > > > > >
> > > > > > > > Sorry for the delayed response. Thank you for including my feedback in the updated version of the manuscript!
> > > > > > > >
> > > > > > > > **Regarding the title:** The suggested new title "On Inherent Robustness of Active Vision Systems" is in my opinion even broader than the initial title, because robustness can mean a lot of things in the deep learning context. I would favor a title that is more specific to what the authors did in their study, but I see the difficulty to come up with such a title.
> > > > > > > >
> > > > > > > > I would be ok with the initial title "On Inherent Adversarial Robustness of Active Vision Systems" if the abstract clearly states that especially transfer attacks were considered (similar as in the author response from 09 Nov 2024 at 21:03). In detail, I would suggest to modify the sentence "We conduct a comprehensive robustness analysis across three categories: adversarially crafted inputs, natural adversarial images, and foreground-distorted images." to do so.
> > > > > > > >
> > > > > > > > Overall, I recommend to accept this paper after this minor issue has been resolved.

---

> > > > > > > > > ### Author Response · Authors · 2024-11-17
> > > > > > > > > **Clarifications to address scope and concerns**
> > > > > > > > >
> > > > > > > > > Thank you for the valuable feedback and constructive engagement regarding clarifying the scope of our manuscript. In response to Reviewer zPwx and Action Editor's suggestions, we have made the following changes to ensure the scope of the paper is clearly defined.
> > > > > > > > >
> > > > > > > > > >  As suggested, we have modified the sentence in the abstract to explicitly state that transfer attacks were the focus of our study.
> > > > > > > > >
> > > > > > > > > > On Page 4 before contributions, we have emphasized the scope by highlighting that our study explicitly focuses on transfer attacks in a black-box setup.
> > > > > > > > >
> > > > > > > > > > Additionally, we have included a couple of sentences in the conclusion to suggest future directions, such as exploring white-box attacks and optimized defenses for active vision systems, aligning with broader adversarial robustness research.
> > > > > > > > >
> > > > > > > > > We hope these changes address the concerns and further clarify the scope of our work.

---

> > > > > > > > > > ### Comment · Reviewer_zPwx · 2024-11-17
> > > > > > > > > >
> > > > > > > > > > Thank you! This addresses my concern regarding the scope of this work. I just posted my recommendation.

---

> ### Comment · Reviewer_zPwx · 2024-11-11
>
> **W2 and W3:** Thank you, I still think that there is room for improvement. Please, address the following minor points:
> - Carefully, check the references again. On the first page most references are wrongly formatted. I would suggest to use a pdf-viewer that does not show the boxes around the references.
> - The new indicator function is not rendered properly in the manuscript.
> - Please, carefully check the capitalization of section/paragraph titles.
> - The newly added equation on page 26 needs to be included in a sentence. Also, the function TGR needs to be properly defined. Also, please do not start sentences with a variable (p. 8 and p. 26).

---

> > ### Author Response · Authors · 2024-11-12
> > **Addressing minor points**
> >
> > Thank you for your additional feedback. We have updated the manuscript and addressed each of these points as follows:
> >
> > 1.  **Citations on Page 1**: We have updated the references on Page 1.
> >
> > 2.  **Indicator function rendering**: The new indicator function has been revised and now renders correctly in the manuscript.
> >
> > 3.  **Capitalization in titles**: We have carefully reviewed and standardized capitalization across all section and paragraph titles.
> >
> > 4.  **TGR definition and reference in the main manuscript**: The newly added equation (3) on page 26 is now referred in the main manuscript in **Transfer attacks with TGR** subsection in Section 4.2. We have expanded on the definition of the TGR function in Appendix Section A.1.5 for enhanced clarity. We have rephrased sentences starting with variables on pages 8 and 26 to enhance readability.
> >
> >
> > Thank you once again for your detailed comments. They were very helpful in guiding us to refine the manuscript.

---

> ### Comment · Reviewer_zPwx · 2024-11-11
>
> **W1.2 and W1.3:** Thank you for the additional experiments! This addresses my concern.

---

### Review · Reviewer_MhiE · 2024-10-27

**Summary Of Contributions:**

The authors hypothesize that dorsal/ventral image classification algorithms are likely robust to adversarial inputs. They then test this hypothesis with previously published GFNet and FALcon architectures.

**Audience:**

Yes

**Broader Impact Concerns:**

Adversarial inputs to images have been thought of as methods to preserve anonymity for people that do not want to be subject to automated classification or detection, particularly for surveillance purposes. Here you are providing algorithms to subvert these protections.

**Claims And Evidence:**

Yes

**Requested Changes:**

There’s too much text in Figure 1. If the figure legend is not sufficient to explain what the diagrams show, the message needs to be pared down and refined. Just something simple: Dorsal stream = localize object, ventral stream = classifies object. As a computer vision engineer without extensive neuroscience background, I have to keep referring back to understand this work.

Figure 5 needs to be more digestible by the reader. There’s a lot of acronyms, and it’s unclear what each of the models and datasets are.

In Figure 5b,c, the figure legends are on top of the result bars. Do these have to be bars? Can it just be a scatter plot? This would be similar to 5a, but without the lines

For Table 1 and Table 2, it would be great to compare to other robustness approaches, many of which are cited at the bottom of page 2.

Figure 11a’s color palate and grid is inconsistent with other paper figures.

It’d be great to have some visualization of what Large Genometric Vicinity and Token-Grandient Regularization attacks look like, potentially in Figure 5.

“For FALcon we employ VGG16 Simonyan & Zisserman (2015) as the dorsal f_{D} stream…” - Why VGG? This is not the best architecture.

What is the MIM attack? I don’t see it defined.

Figure 7 needs more of a figure legend–”Predictions of predictions” is not sufficient.

The fixation map analysis is interesting–aren’t these related to the Saccadic Points defined in Figure 1? It feels like that vocabulary word stopped being used.

Figure 10 is superfluous–why is this figure needed to understand subsequent analysis?

Again, figure 11 would benefit from any other algorithm that has been proposed to be robust to adversarial inputs.

Editorial Changes

Slight nit: I think the vocabulary words: foveation, saccades, dorsal, and ventral can be introduced and defined in the introduction, not the abstract. (Not critical though)

**Strengths And Weaknesses:**

Strengths

Description of multiple adversarial inputs is good.

I generally think the hypothesis is good and interesting and worth exploring.

The fixation map analysis is interesting and highlights the hypothesis being tested with these model architectures.

Weaknesses

While I feel like the description of FalCON is useful for the reader, Figure 3 is essentially what is presented in Ibrayev et al, 2023. Not only did it create this method, it simultaneously discusses the link to foveation and saccade. The same is true for GFNet (Wang et al, 2020). Moreover, I think some description of FalCON and GFNet are helpful, but they are too long for a paper which should largely be novel contribution by you.

There’s a number of acronyms or datasets that are never defined. Are there citations for Swin2 or RVT? I think some definition or intuition on PiT-B, ViT-B/16 or CaiT-S/24 would be helpful. For those in the field, it may be obvious, but in the future, these baselines will surely change.

I think table 2 would benefit from at least one comparison to a passive classifier that is robust to adversarial inputs, like those outlined by the author at the bottom of page 2.

There are just a lot of model architectures floating around, and it’s unclear why each were chosen for the respective task: ResNet34, ResNet50, VGG, VIT, CAIT, PIT.

In Section 4.3, “Hence downsampling the image, distorts the noise along with it, thereby reducing its overall impact on predictions.” Is there any reason why images are downsampled for GFNet except to be computationally efficient? This downsampling step doesn’t seem necessary for Dorsal & Ventral processing, and is just an artifact of GFNet. Moreover, why isn’t the adversarial component just applied to the downsampled images?

In Figure 8, as a human, I don’t even know what item (3) is supposed to be classified as. This is the same for Figure 9. Generally, this analysis is very bespoke and difficult to assess how generalizable it is.

This paper is also quite long and could be pared down substantially, or items moved to the supplement.

---

> ### Author Response · Authors · 2024-11-09
> **Rebuttal for Reviewer MhiE (Addressing Weakness)**
>
> We thank Reviewer MhiE for their encouraging feedback and for highlighting the value of our hypothesis, our analysis of adversarial inputs, and the fixation map results in supporting our tested hypothesis. We have addressed the requested changes regarding figure clarity, citations, and architectural clarifications in the updated version of the manuscript. Thank you again for your valuable feedback.
>
> ### Addressing Weakness
>
> > **Clarifications for Vision Transformer variants** - Citations for target transformer variants such as Swin Transformer V2 (SwinV2) [1] and Robust Vision Transformer (RVT) [2] are provided in the subsection **"Transfer Attacks with Token-Gradient Regularization (TGR) Setup"** under Section 4.2. We have included descriptions for the surrogate transformer variants, such as PiT-B, ViT-B/16, and CaiT-S/24, in the Appendix (A.1.6) to clarify these terms and support readers less familiar with these baselines. Thanks for the feedback.
>
> >> In the subsection **Transfer Attacks with Token-Gradient Regularization (TGR)** in Section 4.2, we follow the TGR setup outlined by Zhang et al. [3], using the same three surrogate transformer architectures (PiT-B, ViT-B/16, and CaiT-S/24) for consistency. Following their approach, our analysis is conducted on 1,000 randomly selected samples from the ImageNet validation set.
>
>
> > **CNN Model Architectures** - In Section 4.1, under **Implementation Details**, we provide the architectural specifics for both active vision systems, GFNet and FALcon.  VGG16 is the default pre-trained architecture used for the FALcon dorsal stream in our analysis. This choice was inherited from the original setup, rather than being specifically selected by us for this study. ResNet34 was selected to generate adversarial samples as it belongs to the same ResNet family as the ventral stream model, allowing us to analyze transfer effects within a similar architecture (ResNet34 versus ResNet50). ResNet50 was chosen to match the exact architecture of the ventral stream. VGG16 was used in Table 2 to evaluate attacks generated with the same architecture as the dorsal stream in FALcon.
>
> > >We have also included results transferred from surrogate DenseNet121 (as discussed in our response to Reviewer zPwx) and observed consistent trends. These additional results are provided in the Appendix A.1.1.
>
>
> > **Attack applied to downsampled images** – Thank you for the insightful comments. The downsampling step in GFNet is indeed an implementation feature for computational efficiency. In Section 4.3, we discuss how downsampling may affect adversarial samples, as it can distort noise and reduce its impact on predictions. To further address your point, we highlighted Setting 3 in Section 4.3, where adversarial attacks are generated on downsampled images. This setup confirms that lower-resolution inputs can increase attack potency. However, in a black-box transfer attack setup, the attacker typically wouldn’t know the input dimensions, which limits the optimization of attacks based on downsampled resolutions. This experiment provides valuable insight into the effects of generating adversarial samples at lower resolutions and addresses your question within the scope of our analysis.
>
> > **Occlusion maps** - In Figure 8, the third sample represents a goldfish from the ImageNet test set, which we recognize is challenging to classify, even without adversarial noise. Despite this, active vision correctly identifies the object in the absence of adversarial perturbations. Figure 8 further shows that, when adversarial noise is added, the dorsal stream’s attention shifts to the background, leading to a misclassification. In Figure 9, we illustrate the varied localization predictions across different fixation points. For the second sample, adversarial noise disrupts most fixation points, resulting in misprediction. Similar visualizations are now provided under Appendix **A.1.8** for TGR attack.
>
> > **Active Vision Models description-** We understand the reviewer’s perspective. As outlined in Section 3, our goal was to provide a focused overview of these methods, emphasizing the key features that contribute to their robustness properties. We have already highlighted these properties in bold within the text to aid clarity. The inference process and any predictions for FALcon are especially important, as we build upon this concept later in the fixation point map analysis. Ensuring clarity in these foundational methods is important to keep the paper self-contained, as Reviewer zPwx noted. Given that our main contributions extend directly from the inference of these methods, we aimed to present them in a focused manner.

---

> > ### Author Response · Authors · 2024-11-09
> > **Rebuttal for Reviewer MhiE (Requested Changes)**
> >
> > ## Addressing Requested Changes
> >
> > > In Figure 1 (c), we mention that the dorsal stream ($f_{D}$) localizes and the ventral stream ($f_{V}$) classifies the object within the localized area. We have highlighted this in the caption for the Figure. Thanks for the suggestion.
> >
> > > **Comparison with Robustness Approaches for Table 1** – **CutMix** [4] is a data augmentation technique that combines images and their labels by randomly cutting and pasting regions between training images, encouraging the model to learn from mixed and occluded objects, thereby improving robustness. In Section 4.2, Table 1, we have included CutMix as a ventral-only method and demonstrated the improved robustness benefits gained by adding FALcon’s dorsal stream against various iterative attacks. In the updated manuscript, we have included Adv-T [5] in Table 1 as an upper-bound baseline—not for direct comparison, but as a performance reference for a method specifically optimized for adversarial defense. In contrast, our model demonstrates significant improvements on adversarially crafted samples over passive-only ventral methods that are not catered towards adversarial defense. We have added a discussion in Appendix A.1.2 to discuss the role of an adversarially trained model as a baseline in our analysis.  [4] and [5] are two approaches that we have mentioned at the bottom of Page 2.
> >
> > >> For transfer attacks using LGV, as shown in **Figure 5(a)**, we again included both CutMix and adversarial training (Adv-T) [5] as additional baselines. However, unlike FALcon-CutMix, FALcon-Adv-T showed limited improvement, as Adv-T is already optimized for worst-case perturbations—a point we emphasize and highlight in under **Transfer Attacks with Large Geometric Vicinity (LGV)** on Page 10 in dark violet.
> >
> > > **Comparison with Robustness Approaches for Figure 11** - In Section 5, for natural adversarial images, we provided a bar plot in Figure 11 where CutMix [4], and Adv-T [5] and RVT [2] are used as ventral-only baselines, with the dorsal stream further enhancing robustness. This is highlighted in this section. For naturally adversarial samples, we observe an improvement even when using an Adv-T-based ventral model, unlike with adversarially crafted samples. This observation is highlighted in the Key Takeaway on Page 17 for Section 5. We have moved the qualitative results that track this quantitative results to Appendix A.2. Since this section highlights results on the ImageNet-A benchmark for naturally adversarial samples, we have provided separate color palate.
> >
> >  > **Fixation Points and Saccadic Points** – Fixation points and saccadic points refer to the same concept in our analysis. To streamline terminology, we clarified in Section 3: "For the remainder of this manuscript, we will refer to saccadic points as fixation points." This terminology is used consistently throughout the manuscript to avoid confusion.
> >
> > > **MIM Attack** – MIM refers to the MIFGSM (Momentum Iterative Fast Gradient Sign Method) attack. We have made this consistent in the manuscript.
> >
> > >  **Figure 7 caption** -  We have expanded on the caption for Figure 7.
> >
> > >**Purpose of Figure 10** – The qualitative results in Figure 10 illustrate why ImageNet-A is particularly challenging, as it includes complex backgrounds, occlusions, and other factors that hinder the performance of passive ventral-only methods. We introduce this motivation in the paper, noting that even humans often need multiple views for accurate interpretation in complex scenes. The explanation is provided in the subsection titled **Catastrophic Mispredictions by Passive Classifiers**, where visualized samples emphasize the distinct challenges of ImageNet-A as opposed to the standard ImageNet benchmark. This dataset is included specifically to assess performance on difficult samples, helping to highlight performance differences for passive classifiers in challenging contexts.
> >
> > > **LGV and TGR** -  We have added sections A.1.4 and A.1.5 to explain the intuition behind the transfer attacks, including Large Geometric Vicinity for CNNs and Token Gradient Regularization for Transformers. We have added Figures 14 and 15 as visualizations with TGR and mentioned under **Appendix A.1.8.**

---

> > > ### Author Response · Authors · 2024-11-09
> > > **Clarifications on Figure 5**
> > >
> > > > **Clarifications on Figure 5** - The caption in Figure 5 explains the results presented in (a), (b), and (c). Analysis is conducted on randomly sampled images from ImageNet, following the methodology outlined in each referenced paper (LGV and TGR). No additional datasets are used in this analysis in Section 4.
> > > >> In Figure 5 (a), we analyze LGV transfer attacks on target CNN-based ventral streams as well as on active systems with both CNN-based dorsal and ventral streams. Details are specified under **Implementation Details** and **Iterative Attacks**. The surrogate model used here is ResNet50, as noted in the caption for 5(a).
> > >
> > > >> In 5 (b) and 5 (c) we analyze results based of TGR attack. Each subplot uses three surrogate transformer architectures—ViT-B/16, PiT-B, and CaiT-S24—as indicated in the legends. The target ventral model for each subplot is given in the title. The passive setup is the ventral model alone, while the active system combines the ventral model with FALcon’s dorsal stream to form a dual-stream configuration. For the **ResNet50 TGR performance**, GFNet is also included as an active vision system sharing the same supervised ventral stream, providing an additional comparison.  Subplot (b) presents the effect of TGR attacks generated with these surrogate transformer architectures on both active and passive systems featuring **CNN-based dorsal and ventral streams.**
> > >
> > > >> For each subplot in c, the target architectures are vision transformer variants. The details of each are mentioned in Appendix A.1.7. titled as **"Target models for Token Gradient Regularization (TGR)"**.  The target active systems in this case is a CNN dorsal and a transformer ventral.  This is highlighted in the results subsection.
> > >
> > >
> > > 1. Ze Liu, Han Hu, Yutong Lin, Zhuliang Yao, Zhenda Xie, Yixuan Wei, Jia Ning, Yue Cao, Zheng Zhang, Li Dong, Furu Wei, and Baining Guo. Swin transformer v2: Scaling up capacity and resolution. In CVPR, 2022.
> > > 2. Xiaofeng Mao, Gege Qi, Yuefeng Chen, Xiaodan Li, Ranjie Duan, Shaokai Ye, Yuan He, and Hui Xue. Towards robust vision transformer. CVPR, 2021.
> > > 3. ianping Zhang, Yizhan Huang, Weibin Wu, and Michael R Lyu. Transferable adversarial attacks on vision transformers with token gradient regularization. In CVPR, 2023.
> > > 4. S. Yun, D. Han, S. Chun, S. Oh, Y. Yoo, and J. Choe. Cutmix: Regularization strategy to train strong classifiers with localizable features. In 2019 IEEE/CVF International Conference on Computer Vision (ICCV), 2019.
> > > 5. Aleksander Madry, Aleksandar Makelov, Ludwig Schmidt, Dimitris Tsipras, and Adrian Vladu. Towards deep learning models resistant to adversarial attacks. In ICLR, 2018.

---

### Decision · Action_Editor_Vury · 2024-12-06

**Recommendation:** Accept as is

**Comment:**

The paper investigates whether an active vision based system, abstractly inspired by the human visual processing stream, is more robust against black box adversarial transfer attacks (i.e., the adversarial inputs were not crafted with gradient access to the model under test). The paper finds that this is indeed the case, which may be due to at least two main aspects: one, the model uses multiple "fixation points", and discards information outside these fixation points, and two, the model uses downsampled patches at these fixation points. Downsampling can be interpreted as a low-pass filter operation, which can be effective against many adversarial attacks that use high-frequency patterns, and similarly the sequential nature of the active vision model is not considered explicitly by most adversarial attacks. Accordingly, the paper finds good empirical results against a number of different black-box transfer attacks. Whether there would be any gains against attacks with gradient access that specifically target the sequential active vision system, remains an important open question (that is beyond the focus of this paper).

I agree with the reviewers that the paper is ready for publication at TMLR and that it is interesting for the wider community - ideally also to spark more work on the aspects that the paper does not answer. The paper is well written and includes a good description of the two main prior methods and the adversarial attacks used, as well as an interesting discussion of insights from human neuroscience that are relevant for computer vision, both of which add value to the community beyond the specific findings.

**Audience:**

All reviewers agree, and I concur, that the paper is interesting to parts of the TMLR audience.

**Claims And Evidence:**

After the author discussion and multiple updates to the manuscript, all reviewers agree that the paper makes falsifiable claims that are supported by accurate and clear evidence. There were some concerns regarding the scope of the initial version as suggested by the title and abstract, and I agree, that it must be made clear that the paper investigates black-box transfer attacks only.